# Functionally distinct cancer-associated fibroblast subpopulations establish a tumor promoting environment in squamous cell carcinoma

Sabrina Schütz [1], Llorenç Solé-Boldo [1], Carlota Lucena-Porcel[2,3], Jochen Hoffmann[4], Alexander Brobeil[2,3], Anke S. Lonsdorf[4], Manuel Rodríguez-Paredes [1,5] ✉ & Frank Lyko [1,5] ✉

Cutaneous squamous cell carcinoma (cSCC) is a serious public health problem due to its high incidence and metastatic potential. It may progress from actinic keratosis (AK), a precancerous lesion, or the in situ carcinoma, Bowen's disease (BD). During this progression, malignant keratinocytes activate dermal fibroblasts into tumor promoting cancer-associated fibroblasts (CAFs), whose origin and emergence remain largely unknown. Here, we generate and analyze >115,000 single-cell transcriptomes from healthy skin, BD and cSCC of male donors. Our results reveal immunoregulatory and matrix-remodeling CAF subtypes that may derive from pro-inflammatory and mesenchymal fibroblasts, respectively. These CAF subtypes are largely absent in AK and interact with different cell types to establish a pro-tumorigenic microenvironment. These findings are cSCC-specific and could not be recapitulated in basal cell carcinomas. Our study provides important insights into the potential origin and functionalities of dermal CAFs that will be highly beneficial for the specific targeting of the cSCC microenvironment.

The skin is composed of two main layers: the outer epidermis, which is a stratified squamous epithelium containing mostly keratinocytes, and the inner dermis, which is crucial for the organ's elasticity and resistance, and consists of fibroblasts as the main cell type[1]. Apart from its sensory role, the skin fulfills important functions such as regulating the body's temperature and water content, producing vitamin D, and defending against external pathogens[2]. As the outermost layer, the epidermis is also an essential barrier against solar ultraviolet radiation (UVR), which can be divided into UVA (315–400 nm) and UVB (280–315 nm)[3]. Although UVB radiation has low skin penetrance, it can cause DNA damage in the form of photoproducts like pyrimidine

dimers in keratinocytes[4]. The cells must repair these photoproducts, since they can interfere with processes such as replication or transcription, ultimately leading to genetic alterations[5]. However, these lesions are generated in large amounts[5], and the accumulation of DNA damage caused by chronic UVB exposure is the main hallmark of keratinocyte carcinomas (KCs), which are the most common cancers in the fair-skinned population[6]. UVA radiation, which has lower energy but greater skin penetrance, also contributes to tumorigenesis, but more indirectly, via oxidative damage. It induces the production of reactive oxygen species (ROS) in the dermis and can cause the degradation of extracellular matrix (ECM) components[7].

[1]Division of Epigenetics, DKFZ-ZMBH Alliance, German Cancer Research Center, 69120 Heidelberg, Germany. [2]Institute of Pathology, Ruprecht-Karls University of Heidelberg, 69120 Heidelberg, Germany. [3]Tissue Bank of the National Center for Tumor Diseases (NCT), 69120 Heidelberg, Germany. [4]Department of Dermatology, University Hospital, Ruprecht-Karls University of Heidelberg, 69120 Heidelberg, Germany. [5]These authors contributed equally: Manuel Rodríguez-Paredes, Frank Lyko. ✉e-mail: m.rodriguez@dkfz.de; f.lyko@dkfz.de

With an increasing global lifetime risk of 9–14% for men and 4–9% for women, cutaneous squamous cell carcinoma (cSCC) accounts for approximately 20% of KCs, second only to basal cell carcinoma (BCC)[8]. This represents at least 700,000 new cases of cSCC per year in the United States alone, which is ~7–8 times as much as malignant melanoma, a skin cancer type arising from melanocytes[9]. Although the metastatic rate of primary cSCC appears relatively low (3–7%)[6], recurrence is often high and increases these numbers to 25–30%[10]. Since the metastatic tumors have a poor prognosis (50-83% and <40% 5-year survival rates for regional and distant metastases, respectively)[11,12], the large numbers of diagnosed cSCC cases not only result in a substantial economic burden for healthcare systems, but also in an absolute number of mortalities that is about half compared to melanoma[13,14]. Importantly, although cSCC can arise directly in the epidermis, it may progress from actinic keratosis (AK), a premalignant lesion, or from the noninvasive carcinoma Bowen's disease (BD, cSCC in situ)[15]. With estimated progression rates of 0.025–16% and 3–5% per year and event, respectively[10,16], these lesions often develop in the same chronically photodamaged regions and share important genetic defects even with the phenotypically normal epidermis, thus making it difficult to ascertain the causes of such progression[17].

In the course of their formation, progression and dissemination, cancer cells establish interactions with the different cell types of the surrounding stroma, the tumor microenvironment (TME), which are ultimately essential for the development of the disease[18]. In this regard, tumor cells are capable of transforming the tissue-residing fibroblasts, as well as other cell types, into cancer-associated fibroblasts (CAFs) by direct contact or in a paracrine manner[19]. Once activated, CAFs support tumor progression by remodeling the surrounding ECM and secreting various factors that favor different aspects of tumorigenesis, including the ability to resist anticancer therapies[20–22]. The emergence and standardization of modern single-cell transcriptomics have enabled the comprehensive study of various cell types, including CAFs, and CAF heterogeneity in different tumor types[23–26]. Interestingly, among the numerous subtypes defined, the presence of immunoregulatory and ECM-remodeling CAF populations, appears to be recurrent, and was also observed in mouse melanoma[26]. However, little is known about cSCC-associated CAFs. In addition, we and others have recently defined different major fibroblast subtypes according to their position and function in the healthy human dermis[27,28]. Since CAFs in skin tumors appear to originate only from skin-resident fibroblasts[29], an analysis of the relationships between fibroblast subtypes and cSCC-associated CAFs may shed light on the origin of dermal CAFs.

Here, we use single-cell RNA sequencing (scRNA-seq) to study CAFs in the context of the disease continuum, from healthy epidermis to invasive cSCC. Our integrated analysis of >115,000 single-cell transcriptomes from healthy UVR-protected and chronically UVR-exposed skin, BD and cSCC samples reveals the presence of two main CAF subtypes, with expression signatures and associated functions that are compatible with those of the immunoregulatory and ECM-remodeling CAF subtypes observed in other cancers[23,24]. Our results suggest that these subpopulations may mainly originate from pro-inflammatory and mesenchymal fibroblast subtypes, respectively. Importantly, multiplexed RNA fluorescence in situ hybridization (FISH) experiments not only validate these CAFs in independent samples but also suggest that they are largely absent in AK, indicating tumor stage-dependent fibroblast activation. Furthermore, cellular interaction analyses suggest CAF subtype-specific activation of signaling pathways in nonmalignant cell types, leading to the establishment of a pro-tumorigenic TME. Finally, our results do not detect similar CAF subtypes in non-metastasizing BCC, thus providing a therapeutic rationale for the specific targeting of the TME.

## Results

### scRNA-seq reveals four fibroblast subtypes across the cSCC continuum

To obtain a comprehensive overview of the disease progression at the single-cell level, we performed a scRNA-seq study that comprised whole-skin biopsies from three healthy, but chronically UVR-exposed skin samples, three BD and five invasive cSCC. We also integrated the datasets of three healthy UVR-protected skin samples of similar age that we had generated for a previous study[27]. The newly generated single-cell transcriptomes were obtained using the 10X Genomics platform and all 14 samples were obtained from male donors, as this is the gender most affected by KCs (Supplementary Table 1).

In total, 115,053 individual transcriptomes passed quality controls (see Methods for details) and formed 30 clusters, visualized in an uniform manifold approximation and projection (UMAP) plot based on their distinct gene expression profiles (Fig. 1a and Supplementary Data 1). Using known marker genes from the literature, these clusters could be assigned to 12 different cell types that are known to be present in human skin[27] (Supplementary Fig. 1a). Importantly, all clusters were detected in all analyzed skin conditions (Supplementary Fig. 1b) and the 14 samples overall contributed to each identified cell type, with an increase in immune cells observed for BD and cSCC samples (Supplementary Fig. 1c). This is consistent with a previous scRNA-seq study of cSCC, in which most immune cells were also isolated from tumor samples[30]. In addition, cell type proportions detected in the six healthy samples were found to be similar to the scRNA-seq dataset of healthy human skin from Tabib et al. [31], with minor differences likely due to the different biopsy collection sites, which can influence skin thickness and composition (Supplementary Fig. 1d).

The disease progression within the integrated dataset could be observed by the increasing number of copy number variation (CNV) events[29,32] that were inferred from the scRNA-seq keratinocyte data (Supplementary Fig. 2a). Furthermore, on average, 92% of the keratinocytes were found to be aneuploid (Supplementary Fig. 2b). Gene Set Enrichment Analyses (GSEA) further confirmed cancer-associated features in BD/cSCC samples. This includes the activation of JAK/STAT and Hedgehog signaling in BD keratinocytes, as well as an enrichment of oxidative phosphorylation and DNA repair in cSCC keratinocytes[33,34] (Supplementary Figs. 2c and d). Moreover, Gene Ontology (GO) analyses on the genes upregulated in BD or cSCC keratinocytes compared to healthy keratinocytes indicated an enrichment of tumorigenic processes like angiogenesis, transcription upregulation and immune responses for BD, as well as aerobic respiration and response to oxidative stress for cSCC (Supplementary Fig. 2e). Finally, an increased expression of proliferation-related genes that are important during the G2/M cell cycle phase[29] could be observed for keratinocytes from cSCC samples (Supplementary Figure 2f), which also showed enriched expression of a previously defined cSCC-derived cancer cell gene signature[30] (Supplementary Fig. 2g). Taken together, these findings strongly suggest that keratinocytes from BD and cSCC are indeed cancer-derived keratinocytes.

As expected, all non malignant cells clustered by cell type, but tumor keratinocytes did not cluster separately from their healthy counterparts, or formed clusters based on different patients and/or disease status. This can be explained by the integrated workflow that was used for this analysis, which ensures that nonmalignant cells from the same cell type and present in different datasets cluster together (see Methods for details). However, simple merging without integration of scRNA-seq data from all samples revealed entity-specific keratinocyte clusters (Supplementary Fig. 2h) and even separated patient-specific keratinocytes from cSCC samples (Supplementary Fig. 2i), as it was shown in previous scRNA-seq studies of skin tumors[29].

Based on the expression of the canonical markers *LUM, DCN, VIM, PDGFRA* and *COL1A2*[35], a total of 27,382 cells within four clusters were classified as fibroblasts. Analyses of the most representative genes

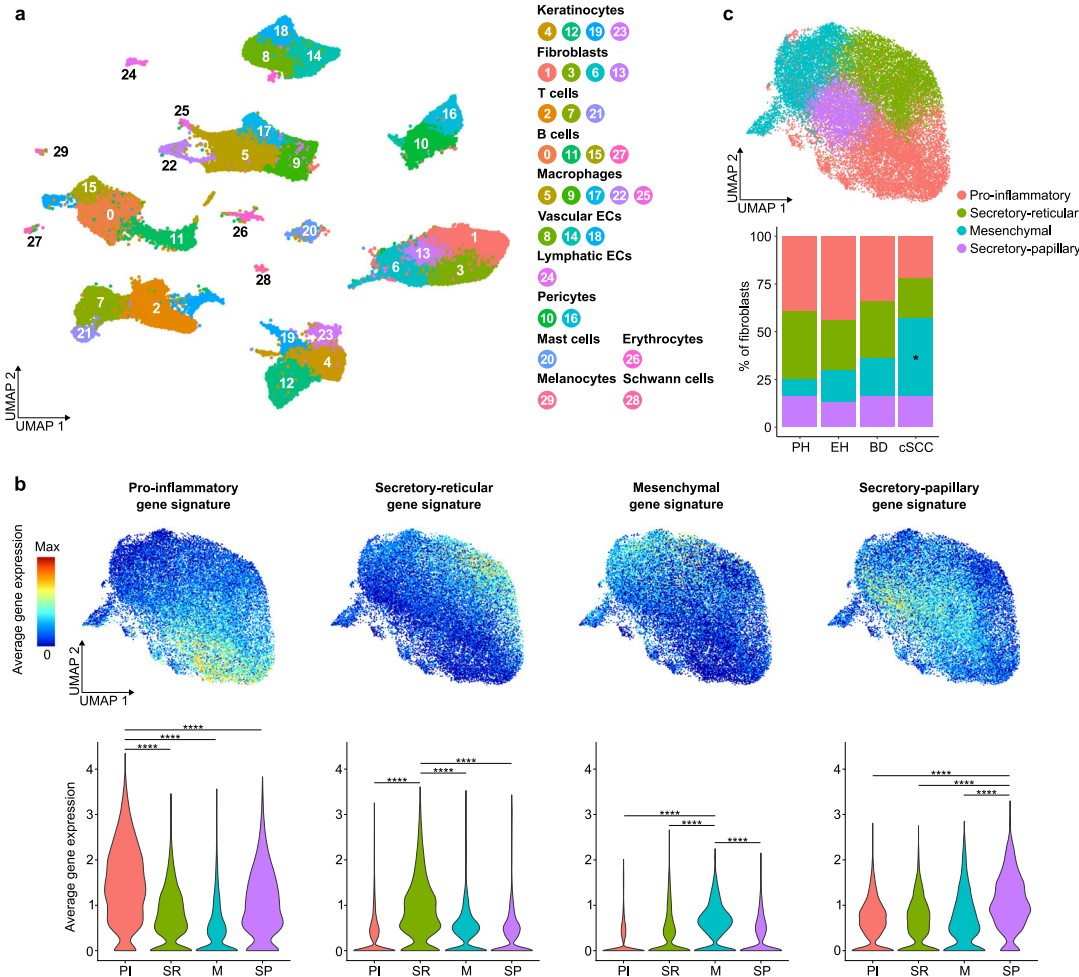

**Fig. 1 | Integrated scRNA-seq analysis defines fibroblast subpopulations along the cSCC disease continuum. a** UMAP plot visualizing cell types identified in UVR-protected healthy skin ($n = 3$ patients), chronically UVR-exposed healthy skin ($n = 3$ patients), BD ($n = 3$ patients) and cSCC ($n = 5$ patients) samples. Each dot represents a single cell ($n = 115,053$ cells). Coloring is according to the unsupervised clustering performed by Seurat v4. **b** UMAP plots and violin plots depicting the average gene expression of known gene signatures for the four main fibroblast subpopulations previously defined in UVR-protected healthy human skin[27]. For UMAP plots, red indicates maximum gene expression, while blue indicates low or no expression. For violin plots, statistical analyses were performed using a two-sided Wilcoxon Rank Sum test ($p$ values for all indicated comparisons <2.2e-16,

****: $p$ value < 0.0001). **c** UMAP plot showing the four fibroblast clusters that were detected in all sample entities, as well as their different proportions along cSCC development (bar plot). For the latter graph, a one-way analysis of variance (ANOVA) was first applied for each fibroblast subpopulation to assess potential changes in their proportions between the four sample types. Pairwise comparisons using the Holm-Sidak test (*: $p$ value < 0.05) were then performed for each fibroblast subpopulation to reveal the specific sample types between which a statistically significant change of proportions was occurring. UVR: ultraviolet radiation, PH: UVR-protected healthy, EH UVR-exposed healthy, BD Bowen's disease, cSCC cutaneous squamous cell carcinoma, PI pro-inflammatory, SR secretory-reticular, M mesenchymal, SP secretory-papillary. Source data are provided as a Source Data file.

expressed in each of these clusters (Supplementary Data 2) revealed that they represent the four fibroblast subpopulations that we had previously defined in the healthy, UVR-protected human skin, and which we named according to their different functions and dermal location: pro-inflammatory, secretory-papillary, secretory-reticular and mesenchymal[27] (Fig. 1b and Supplementary Fig. 3a). This was further supported by GO analyses using the same four gene sets (Supplementary Fig. 3b). Most interestingly, although the four subpopulations could be observed during the entire cSCC disease continuum, their proportions with respect to the total number of cells per sample type changed with the disease progression (Fig. 1c and Supplementary Table 2). Thus, while slightly fewer pro-inflammatory and secretory-reticular fibroblasts were detected in cSCC tumors, the number of mesenchymal fibroblasts was significantly increased compared to healthy, UVR-protected skin ($p$ value < 0.05 for both ANOVA and subsequent Holm-Sidak tests for pairwise comparisons). Although mesenchymal fibroblasts from cSCC samples showed slightly higher expression of genes related to the S phase of the cell cycle compared

to UVR-protected skin (Supplementary Fig. 3c), additional cell cycle and proliferation analyses could not attribute these changes to a significantly higher proliferation rate (Supplementary Figs. 3d and 3e). The observed progression-related changes in fibroblast heterogeneity also raised the possibility that CAFs originate from specific fibroblast subpopulations.

## Detection of two major CAF subtypes with different functions

To identify cSCC-associated CAFs, we focused on the analysis of fibroblasts from BD and cSCC samples. Trajectory inference suggested a progression from healthy fibroblasts towards BD- and cSCC-derived fibroblasts (Supplementary Fig. 4a). Furthermore, fibroblasts of both entities showed an increased expression of the known CAF-related marker genes *FAP* and *ACTA2*[19] (Supplementary Fig. 4b). Isolation of these cells and second-level clustering revealed two main CAF populations with different gene expression profiles (Fig. 2a and Supplementary Data 3). GO analyses showed terms related to inflammation processes, such as cytokine-mediated signaling and inflammatory

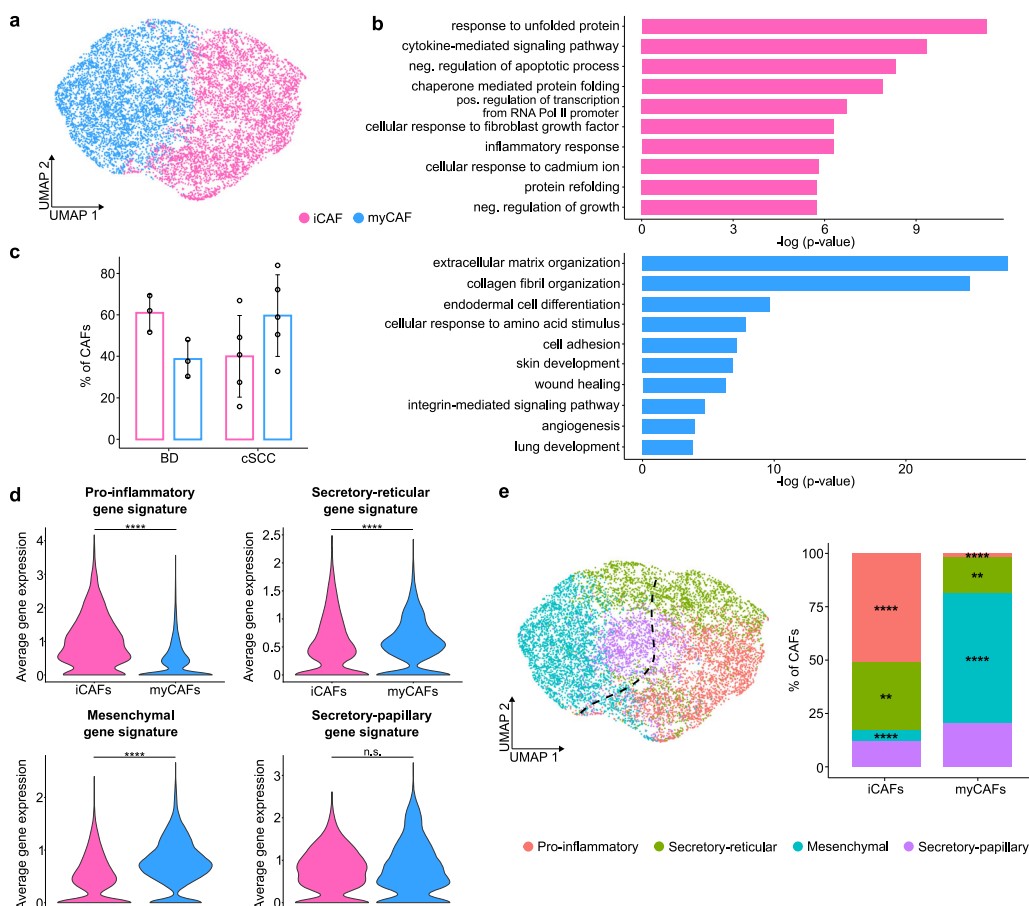

**Fig. 2 | BD and cSCC show two major CAF subtypes with different functions and a probable origin in distinct dermal fibroblasts. a** UMAP plot visualizing the two main CAF populations. **b** Top 10 terms from a Gene Ontology (GO) analysis with the most representative expressed genes of each CAF population. Terms are ordered according to *p*-value, determined by the Fisher's Exact test. **c** Bar plot depicting the average proportions of cutaneous iCAFs and myCAFs in BD and cSCC. Bars indicate mean values and each dot represents an analyzed scRNA-seq sample (BD: *n* = 3 patients, cSCC: *n* = 5 patients). Statistical analyses of the differences were performed using paired/unpaired *t*-tests and error bars represent the standard deviation. **d** Violin plots showing the average gene expression of the known gene signatures corresponding to the four previously defined fibroblast subpopulations[27] in cutaneous iCAFs and myCAFs. Statistical analyses were performed using a two-sided Wilcoxon Rank Sum test (pro-inflammatory: *p* value < 2.2e-16, secretory-reticular: *p* value = 1.506e-09, mesenchymal: *p* value < 2.2e-16,

secretory-papillary: *p* value = 0.4435, ****: *p* value < 0.0001). **e** Left: UMAP plot depicting the correlation between iCAFs and myCAFs (as shown in **a**) and the four fibroblast subpopulations already present in healthy skin and identified in our original integrated scRNA-seq analysis (see Fig. 1). Right: Bar plot with the proportion of cutaneous iCAFs and myCAFs related to secretory-papillary, secretory-reticular, pro-inflammatory and mesenchymal fibroblasts. Statistical analyses of the differences were performed using two-sided paired *t*-tests (pro-inflammatory: *p* value = 0.0000111, secretory-reticular: *p* value = 0.00315, mesenchymal: *p* value = 0.000000955, secretory-*p*apillary: *p* value = 0.109, **: *p* value < 0.01, ****: *p* value < 0.0001). iCAF: inflammatory cancer-associated fibroblast, myCAF myofibroblastic cancer-associated fibroblast, BD Bowen's disease, cSCC cutaneous squamous cell carcinoma, n.s.: not significant. Source data are provided as a Source Data file.

response in the one subpopulation, while the other subpopulation was defined by terms related to ECM organization and collagen fibrils (Fig. 2b). Similar CAF subpopulations have also been detected in other cancer types, such as pancreatic ductal adenocarcinoma (PDAC), where they were termed inflammatory CAFs (iCAFs) and myofibroblastic CAFs (myCAFs)[23,24]. Indeed, comparative gene expression analyses suggested a high degree of resemblance between pancreatic cancer- and cSCC-related CAF subpopulations (Supplementary Fig. 4c), which was corroborated by GSEA (Supplementary Figs. 4d and e). Dermal iCAFs and myCAFs also showed high similarity scores for the respective pancreatic CAF subtypes (Supplementary Fig. 4f). Therefore, the existing nomenclature was adopted.

The proportions of cutaneous iCAFs and myCAFs were subsequently investigated in BD and invasive cSCC. In BD samples, we observed more iCAFs (~62%) than myCAFs (~38%), whereas slightly more myCAFs (~57%) than iCAFs (~43%) could be detected in cSCC cases (Fig. 2c). Although the expression of cell cycle-related genes is

significantly different between iCAFs and myCAFs in BD and cSCC samples (Supplementary Fig. 4g), the small effect size of these differences, as well as additional cell cycle and proliferation analyses (Supplementary Figs. 4h and i), suggest that the potential change in CAF composition during cSCC progression is unlikely to be related to an increased proliferative capacity of iCAFs in BD or myCAFs in cSCC, but rather to a tumor-specific need for the functions provided by each CAF subtype.

To identify the potential cell type of origin of both CAF subtypes, mean expression levels of marker genes from healthy skin fibroblast subpopulations[27] were determined in cutaneous iCAFs and myCAFs. This revealed an increased expression of pro-inflammatory fibroblast gene signatures in iCAFs, and enriched expression of mesenchymal fibroblast gene signatures in myCAFs (Fig. 2d), whereas cSCC-related CAF signatures were not enriched in healthy fibroblasts (Supplementary Fig. 4j). This observation indicated that the expression of the marker genes of the four fibroblast subpopulations was present along

the cSCC continuum. It also suggested that BD- and cSCC-derived fibroblasts specifically showed a CAF-related gene expression pattern that allowed their classification into iCAFs and myCAFs. Moreover, pairwise comparisons performed with the most representative genes of iCAFs or myCAFs, and the four subtypes of dermal fibroblasts detected the largest overlaps between pro-inflammatory fibroblasts and iCAFs (15 genes), and between mesenchymal fibroblasts and myCAFs (22 genes) (Supplementary Fig. 4k). Similar results were obtained in the integrated scRNA-seq analysis with all healthy and diseased skin cells, which showed that cutaneous iCAFs significantly clustered with the pro-inflammatory fibroblasts ($p$ value < 0.0001, paired t-test), while myCAFs significantly clustered with the mesenchymal fibroblasts ($p$ value < 0.0001, paired t-test) (Fig. 2e). Taken together, our observations indicate the existence of two CAF subtypes in cSCC tumorigenesis, one more related to inflammatory processes and the other more related to ECM organization. Our results also suggest that they may originate from distinct dermal fibroblast subtypes.

### cSCC-related CAFs are not yet activated in precancerous AK lesions

To validate the two CAF populations observed in BD and cSCC, and to further address their possible emergence from the different dermal fibroblast subtypes, we used multiplexed RNA FISH on sections from an independent set of tumor samples and healthy chronically UVR-exposed control skin samples (Supplementary Table 3). Representative markers of general cSCC-related CAFs (*COL3A1*), iCAFs (*C3* and *IGF1*) and myCAFs (*MMP11* and *WNT5A*), were selected based on their high expression level and specific expression pattern in our scRNA-seq experiments (Supplementary Data 4). Similarly, pro-inflammatory (*APOE*), secretory-papillary (*APCDD1* and *DIO2*), secretory-reticular (*CCN5/WISP2*) and mesenchymal (*POSTN*) fibroblast-specific genes were selected based on our previous work[27]. Finally, *KRT14* was included as an additional control to detect both tumor regions and healthy basal epidermal keratinocytes[36]. Our results showed a strong signal for the general CAF marker *COL3A1* in all sections from BD and invasive cSCC samples, and a much lower signal intensity in the healthy chronically UVR-exposed skin. Similar results were obtained for the specific iCAF and myCAF marker genes (Fig. 3a and Supplementary Figs. 5a and b). Importantly, these differences could also be observed in non-tumoral but chronically UVR-exposed skin that was present at the edges of the analyzed cSCC tissue sections (Supplementary Figs. 5c and d). Relative to malignant cells, no specific spatial patterns could be observed for iCAFs and myCAFs in BD and cSCC, as both subpopulations were detected throughout the TME (Fig. 3a and Supplementary Figs. 5c and d).

We then quantified iCAFs as *COL3A1*-positive cells that were also positive for *IGF1*, *C3* or both, while we quantified myCAFs as *COL3A1*-positive cells that were also positive for *MMP11*, *WNT5A* or both. Our results showed a significantly increased presence of myCAFs in comparison to iCAFs in both BD (49% vs. 13%, $p$ value < 0.001, paired t-test) and cSCC samples (56% vs. 15%, $p$ value < 0.001, paired t-test), which might suggest an increased presence of myCAFs during cSCC progression, in agreement with the scRNA-seq results (Fig. 3b), and despite the minimal number of markers used by RNA FISH to identify this subpopulation, compared to the single-cell approach. The minor fraction of cells expressing mixed iCAF and myCAF marker combinations were not considered during the quantification, as no clear subtype classification was possible (Supplementary Fig. 6a). Furthermore, since not all iCAFs and myCAFs necessarily express all of their specific marker genes, we also quantified double-positive cells as iCAFs (*COL3A1*⁺, *C3*⁺ or *IGF1*⁺) and myCAFs (*COL3A1*⁺, *MMP11*⁺ or *WNT5A*⁺), respectively (Supplementary Fig. 6b).

Furthermore, RNA FISH analyses showed positive signals for fibroblast subpopulation marker genes in all samples (Fig. 3c and

Supplementary Fig. 7a). Additionally, between 32% (cSCC) and 59% (BD) of myCAFs expressed the mesenchymal fibroblast marker *POSTN*, compared to 22% (cSCC, $p$ value = 0.056, paired t-test) and 30% (BD, $p$ value < 0.001, paired t-test) of iCAFs. Conversely, while 8% (BD) and 10% (cSCC) of iCAFs were positively stained for the pro-inflammatory gene *APOE*, these percentages significantly decreased to 1-2% in myCAFs (BD: $p$ value < 0.05, paired t-test, cSCC: $p$ value < 0.05, Wilcoxon Signed Rank Test) (Fig. 3d).

Interestingly, our RNA FISH assays did not detect a positive *COL3A1* signal in two out of three precancerous AK samples (Fig. 3e). Similar results were obtained for AK regions surrounding cSCC tumors (Supplementary Figure 7b). Moreover, the *COL3A1* positive AK sample did not express any of the iCAF or myCAF markers, suggesting that fibroblasts have not yet been fully activated into CAFs at this stage (Supplementary Fig. 7c). Control stainings showed marker gene expression for the four fibroblast subtypes in all AK samples (Supplementary Fig. 7d). Therefore, our multiplexed RNA FISH assays not only provided important confirmation of the two CAF subtypes defined by our scRNA-seq experiments, but also specified the window of time for the emergence of CAFs.

### CAFs establish a pro-tumorigenic tumor microenvironment

To further investigate the functional roles of cutaneous iCAFs and myCAFs during cSCC tumorigenesis, we used our BD and cSCC scRNA-seq data for computational cell-cell communication analyses with CellChat[37]. A general analysis based on known ligand-receptor interactions and the expression data from BD and cSCC samples identified CAFs as the cell type with the strongest outgoing interaction signals (Fig. 4a). Interestingly, in BD, iCAFs appeared to be more important "sender" cells than myCAFs, and T cells were defined as the cell type with the strongest incoming interactions. In contrast, in cSCC, the strongest outgoing interaction signals were detected in myCAFs, while malignant keratinocytes showed the strongest incoming interactions. Further analysis of cellular communication between all CAFs and cancer cells in the cSCC TME revealed that collagen-related and Fibronectin 1 (FN1) signaling pathway networks are particularly active (Supplementary Fig. 8a). Both CAF populations were observed to secrete mainly type I and type VI collagens as well as FN1, which are all predicted to bind to CD44 and the Syndecans-1 and -4 (SDC1/4) on the surface of cSCC keratinocytes (Supplementary Fig. 8b). CD44 is known to be involved in various cellular pathways, such as PI3K/AKT and Src/MAPK signaling, which contribute to cancer cell proliferation and invasion[38].

In addition, the predicted cell-cell interaction patterns in the cSCC TME supported functional differences between cSCC-related iCAFs and myCAFs. More specifically, cSCC iCAFs showed increased expression of Adrenomedullin (*ADM*) and the C-X-C motif chemokine ligand 12 (*CXCL12*), which are secreted ligands for the Calcitonin-like receptor (CALCRL) on vascular endothelial cells, and the C-X-C motif chemokine receptor 4 (CXCR4) on macrophages and T cells (Fig. 4b and Supplementary Fig. 8c). CAF-derived ADM is known to promote tumor growth by modulating angiogenesis and immune cell chemotaxis[39,40]. The CXCL12-CXCR4 pathway has a prominent role in tumor immunosuppression, also in cSCC[41]. On the other hand, myCAFs showed specific interactions related to the Angiopoietin (ANGPTL) and noncanonical WNT (ncWNT) pathway networks. More specifically, they secrete ANGPTL2 and WNT5A, which can bind to integrin α5β1 (ITGA5/ITGB1) on vascular endothelial cells, as well as to the melanoma cell adhesion molecule (MCAM) expressed by pericytes and vascular endothelial cells (Fig. 4c and Supplementary Fig. 8d). The interaction between ANGPTL2 and integrin α5β1 is known to activate p38 MAPK signaling, which leads to enhanced ECM remodeling[42]. WNT5A-MCAM signaling in pericytes has previously been implicated in increased cell motility[43]. Similar results were also obtained with the ligand-receptor analysis framework LIANA[44], which is based on the consensus of

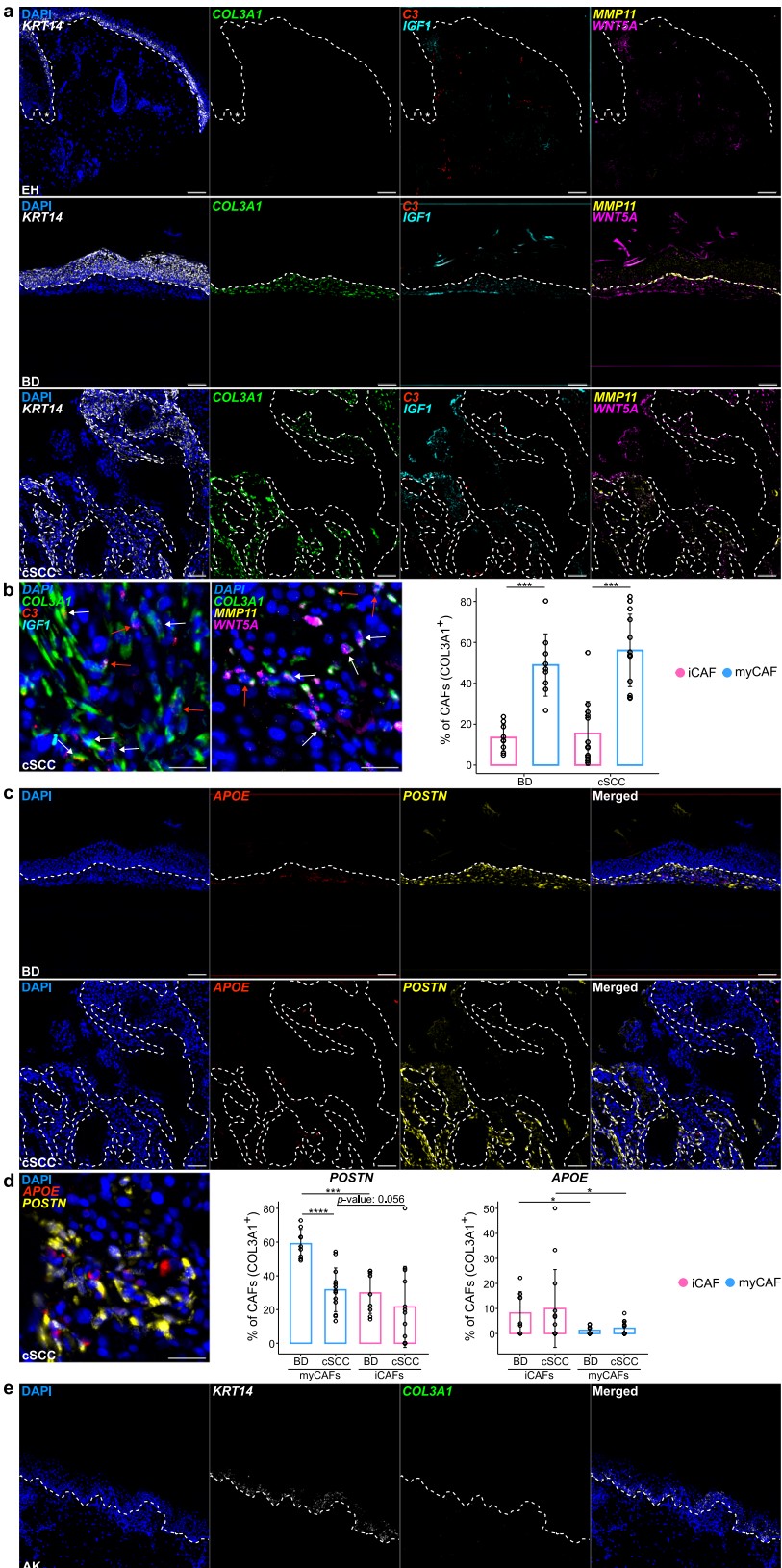

several interaction analysis methods and resources (Supplementary Fig. 8e).

Finally, we also validated the key predicted interactions by multiplexed RNA FISH experiments on formalin-fixed paraffin embedded (FFPE) sections from independent cSCC samples. Although, both CAF subpopulations were detected throughout the TME, an enrichment of

iCAFs (*COL3A1*[+], *C3*[+]) expressing the ligand *CXCL12* was observed in close proximity to T cells (*CD3D*[+]) expressing the receptor *CXCR4* (Fig. 4d and Supplementary Fig. 8f). Moreover, myCAFs showing co-expression of *COL3A1* and *WNT5A* were enriched next to vascular endothelial cells (*VWF*[+]) expressing the mRNA encoding the MCAM receptor (Fig. 4e and Supplementary Fig. 8g). These findings indicate

**Fig. 3 | Multiplexed RNA FISH validates cutaneous CAF subtypes and indicates their absence in AK. a** Representative microscopy images of FFPE sections from chronically UVR-exposed healthy skin, BD and cSCC hybridized with fluorescent probes against the mRNA of *KRT14* (white) as keratinocyte marker, *COL3A1* (green) as general CAF marker, *C3* (red) and *IGF1* (cyan) as iCAF marker genes as well as *MMP11* (yellow) and *WNT5A* (magenta) as myCAF marker genes. Nuclei were counterstained with DAPI. Dashed lines separate epidermal keratinocytes from dermal cells and asterisks mark hair follicles. Scale bar: 100 μm. **b** Detailed images from cSCC tissue in **a** with positive cells for iCAF marker genes *C3* and/or *IGF1* (left) and myCAF marker genes *MMP11* and/or *WNT5A* (center). White arrows indicate double-positive CAFs (*COL3A1*⁺, *C3*⁺ or *IGF1*⁺ for iCAFs; *COL3A1*⁺, *MMP11*⁺ or *WNT5A*⁺ for myCAFs). Red arrows indicate triple-positive CAFs (*COL3A1*⁺, *C3*⁺, *IGF1*⁺ for iCAFs; COL3A1⁺, MMP11⁺, WNT5A⁺ for myCAFs). Scale bar: 25 μm. Quantification of these positive signals in CAFs (*COL3A1*⁺ cells) in BD and cSCC (right). **c** Representative microscopy images of FFPE sections from BD and cSCC hybridized with fluorescent probes against the mRNA of *APOE* (red) as pro-inflammatory fibroblast marker and *POSTN* (yellow) as mesenchymal fibroblast marker. Nuclei were counterstained with DAPI. Dashed lines separate epidermal keratinocytes from dermal cells. Scale bar: 100 μm. **d** Detailed image from cSCC tissue in **c** with

positive cells for pro-inflammatory and mesenchymal marker genes *APOE* and *POSTN*, respectively (left). Scale bar: 25 μm. Quantification of these positive signals in myCAFs (co-expression with *COL3A1* and *MMP11*/*WNT5A*) (center) and iCAFs (co-expression with *COL3A1* and *C3*/*IGF1*) in BD and cSCC (right). **e** Representative microscopy images of a FFPE AK sample ($n = 3$ patients) hybridized with fluorescent probes against the mRNA of *KRT14* (white) as keratinocyte marker and *COL3A1* (green) as general CAF marker. Nuclei were counterstained with DAPI. Dashed lines separate epidermal keratinocytes from dermal cells. For quantifications in **b** (BD: $p$ value = 0.000821, cSCC: $p$ value = 0.000518) and **d** (*POSTN* myCAFs: $p$ value = 0.0000198, *POSTN* BD: $p$ value = 0.000659, *APOE* BD: $p$ value = 0.0321, *APOE* cSCC: $p$ value = 0.049), statistical analyses were performed using two-sided unpaired/paired t-tests (*: $p$ value < 0.05, ***: $p$ value < 0.001, ****: $p$ value < 0.0001) and error bars show the standard deviation. Bars indicate mean values and each dot represents a dermal region with around 100 *COL3A1*⁺ cells/CAFs (BD: $n = 9$ dermal regions, cSCC: $n = 13$ dermal regions). FFPE formalin-fixed paraffin embedded, UVR ultraviolet radiation, EH UVR-exposed healthy skin, AK actinic keratosis, BD Bowen's disease, cSCC cutaneous squamous cell carcinoma, iCAF inflammatory cancer-associated fibroblast, myCAF myofibroblastic cancer-associated fibroblast. Source data are provided as a Source Data file.

that CAF subtype-specific interactions with nonmalignant cell types contribute to establish a pro-tumorigenic cSCC TME.

To gain further insight into the influence of CAFs on nonmalignant cell types, we compared functional GO terms obtained for pericytes and vascular endothelial cells in healthy chronically UVR-exposed skin and cSCC. In cSCC, we observed that both cell types showed a considerable increase in functional annotations for angiogenesis, cell migration and cell adhesion, i.e., more function-related terms and/or substantially decreased $p$ values, which is consistent with a higher number of genes supporting the terms (Figs. 4f and g). Additional GO analyses performed specifically with the genes upregulated during cSCC tumorigenesis also supported these findings (Supplementary Fig. 8h). In summary, our analysis of cell-cell communications during cSCC progression revealed that CAFs play prominent roles in specific signaling networks involved in cancer cell proliferation and invasion. Tumor promoting and subtype-specific CAF functions were also supported by their predicted contributions to immune- or matrix-related signaling networks, respectively, as well as by their specific effects on nonmalignant cells of the TME.

### cSCC-related CAFs differ from BCC CAFs

Finally, in order to investigate whether cSCC-related CAFs are present in the other major KC subtype, we used publicly available scRNA-seq data to determine whether these CAF subtypes could also be detected in BCC[45,46]. The datasets were integrated as before (see Methods for details and Supplementary Fig. 9a) and suggested the presence of a single fibroblast cluster with 3,998 cells (Supplementary Fig. 9b). A comparison of these cells and the 5956 cSCC-related fibroblasts from our scRNA-seq dataset showed a lower expression of both the CAF marker genes from Öhlund et al. [23], and our CAF-related genes (Supplementary Data 4) in BCC-derived fibroblasts (Supplementary Fig. 9c).

Subsequent isolation and unsupervised re-clustering of these fibroblasts resulted in four clusters (Supplementary Fig. 9d and Supplementary Data 5). Only two of them (FB 1 and FB 2) showed high expression levels of both cSCC-related (Fig. 5a and Supplementary Data 4) and classical CAF markers, such as *FAP* and *ACTA2*[19] (Fig. 5b). However, neither of these subpopulations corresponded to the iCAFs and myCAFs observed in the cSCC disease continuum (Supplementary Fig. 9e). To reduce possible confounding factors in this analysis, we next performed second-level clustering using only the 2,223 cells from those two subgroups (Fig. 5c and Supplementary Data 6). This resulted in another four clusters that again could not be designated as iCAFs or myCAFs (Supplementary Fig. 9f). Some of the most representative genes of cSCC-related CAFs, such as *CXCL12*, *JUND*, *TSC22D3* (iCAFs)

and *MMP11*, *TAGLN*, *COL11A1* (myCAFs) could be detected in a single cluster (CAF 2) (Fig. 5d), which suggests a mixed CAF phenotype with GO terms related to translation and ECM organization (Supplementary Fig. 9g). GO analyses of the other clusters revealed involvement in ion homeostasis, angiogenesis and wound healing (Supplementary Fig. 9h). Taken together, our analysis of CAFs in non-metastasizing BCC therefore suggests that these cells differ from cSCC-derived iCAFs and myCAFs.

## Discussion

This work addresses the origin of KC CAFs and the time window for fibroblast activation during KC tumorigenesis, as well as the impact of functionally different CAF subtypes in the TME. For this purpose, we have used single-cell transcriptomics and multiplexed RNA FISH to analyze the cSCC disease continuum. Our integrated analysis also included data from UVR-protected skin samples as a control for healthy skin. All samples were obtained from male individuals, as this group constitutes the classical majority of patients for KCs.

Our study provides a number of interesting findings. First of all, we observed two major types of CAFs during cSCC progression that were validated with multiplexed RNA FISH assays on independent samples. Based on the functions associated with their expression signatures, these cells are likely to play immunoregulatory and ECM remodeling roles, respectively. Several groups have recently characterized CAFs in different types of cancer using scRNA-seq, and detected subtypes with similar putative functions[23–26]. However, a detailed characterization of the origin of this heterogeneous population has been lacking. We have recently used scRNA-seq to spatially and functionally define the four main dermal fibroblasts subtypes in healthy skin[27], which have been validated by other groups[28]. Our data show that cSCC-related iCAFs may mostly arise from the pro-inflammatory fibroblasts, while myCAFs may largely arise from the mesenchymal subtype. Fibroblasts of the mesenchymal subtype have also recently been found to be involved in fibrotic skin diseases, where ECM dysregulation is paramount[28]. Further studies, such as in vivo lineage tracing will be required to verify the origin/s of the defined CAFs.

The time window of fibroblast activation represents another key aspect for the characterization of CAF development. In the mouse, for example, fibroblasts in early PDAC resembled the cells in healthy pancreatic tissue, and CAFs were only detected in later stages[47]. However, in human PDAC, myCAFs were already detected in the noninvasive intraductal papillary mucinous neoplasms (IPMNs), but iCAFs were only present in invasive PDACs[48]. Using multiplexed RNA FISH assays, we were able to narrow down the time window for

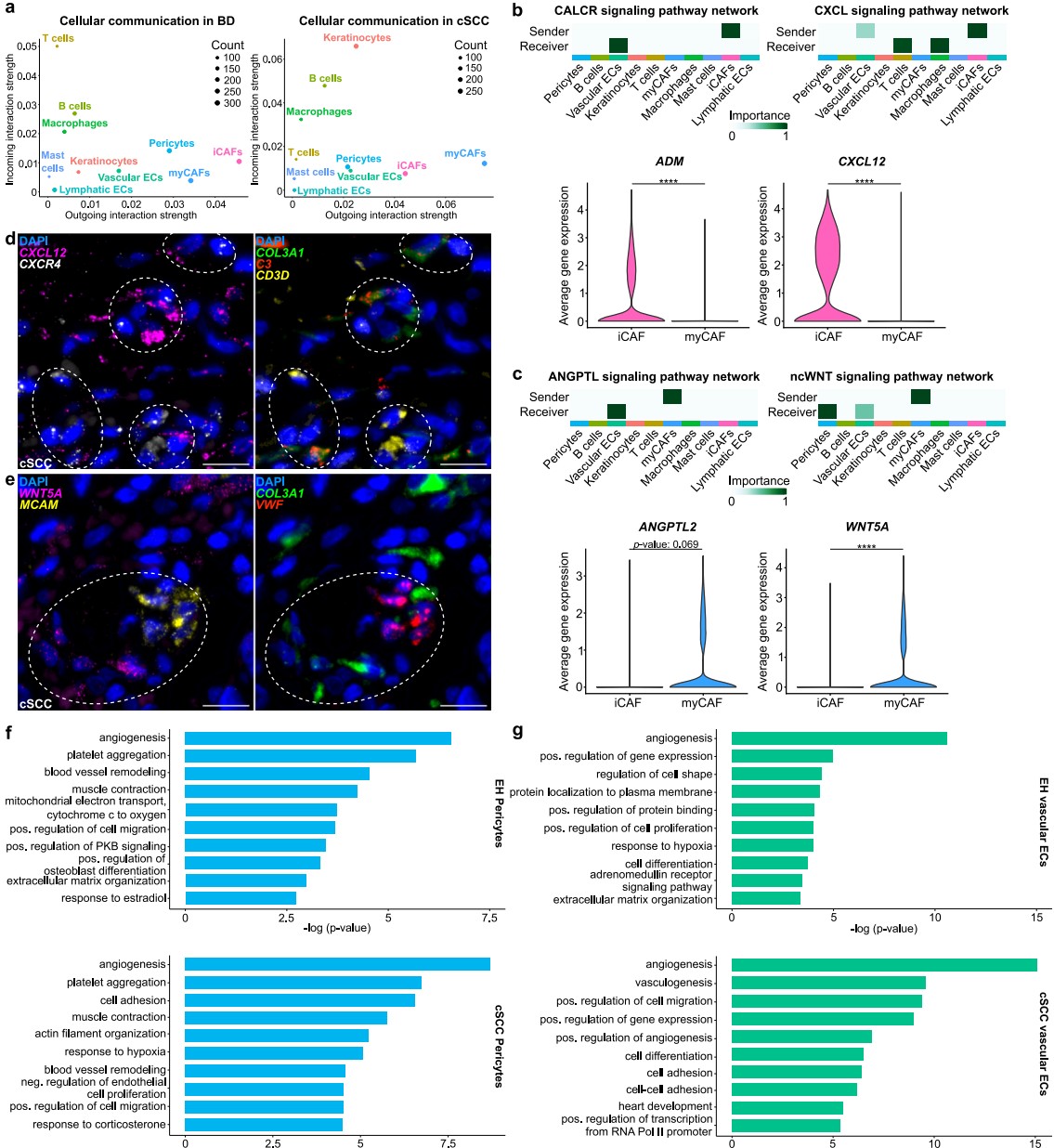

**Fig. 4 | Cutaneous iCAFs and myCAFs interact with different cell types to establish a pro-tumorigenic tumor microenvironment during cSCC development. a** Scatter plots visualizing the outgoing and incoming interaction strength (X-axes and Y-axes, respectively) between the different cell types in BD (left) and cSCC (right) samples. **b** and **c** Upper rows: heatmaps showing the prediction of the most important "sender" and "receiver" cell types within the cSCC tissue for different signaling pathway networks. White or light green correlates with no or low involvement, while dark green represents high importance. Bottom rows: violin plots representing the average gene expression of the most important ligands secreted by CAFs. Statistical analyses were performed using a two-sided Wilcoxon Rank Sum test (*ADM*: *p* value < 2.2e-16, *CXCL12*: *p* value < 2.2e-16, *WNT5A*: *p* value < 2.2e-16, ****: *p* value < 0.0001). **d** Representative microscopy image of a FFPE section from a cSCC sample (*n* = 2 patients) hybridized with fluorescent probes against the mRNA of *CXCL12* (magenta), encoding an iCAF-derived ligand, *CXCR4* (white), encoding a receptor on T cells, *COL3A1* (green) as general CAF marker, *C3*

(red) as an iCAF marker gene and *CD3D* (yellow) as a T cell marker. Nuclei were counterstained with DAPI. Dashed circles highlight interaction areas. Scale bar: 25 μm. **e** Representative microscopy image of a FFPE section from a cSCC sample (*n* = 2 patients) hybridized with fluorescent probes against the mRNA of *WNT5A* (magenta), encoding a myCAF-derived ligand and used as myCAF marker gene, *MCAM* (yellow), encoding a receptor on vascular ECs, *COL3A1* (green) as general CAF marker, and *VWF* (red) as a marker for vascular ECs. Nuclei were counterstained with DAPI. Dashed circles highlight interaction areas. Scale bar: 25 μm. **f** and **g** Top 10 terms from Gene Ontology (GO) analyses with the most representative expressed genes of pericytes (**f**) and vascular ECs (**g**) in chronically UVR-exposed healthy skin and cSCC samples. Terms are ordered according to *p* value, determined by the Fisher's Exact test. EC endothelial cell, iCAF inflammatory cancer-associated fibroblast, myCAF myofibroblastic cancer-associated fibroblast, UVR ultraviolet radiation, EH UVR-exposed healthy, cSCC cutaneous squamous cell carcinoma. Source data are provided as a Source Data file.

fibroblast activation, as we did not detect CAFs in nonhealthy but precancerous tissue. The absence of CAFs in AK is particularly interesting, considering the similarities between AK and invasive cSCC in terms of mutational spectrum[17,49], gene expression profiles[50] or even DNA methylomes[51]. One of the main pathological differences between

the two conditions could lie in their distinct capacity to generate CAFs, which may contribute to the ability of AKs to regress spontaneously[15].

Furthermore, we showed that both CAF subtypes coexist in the TME, consistent with published findings from different cancer entities[23,26,52,53]. However, we did not observe any specific spatial

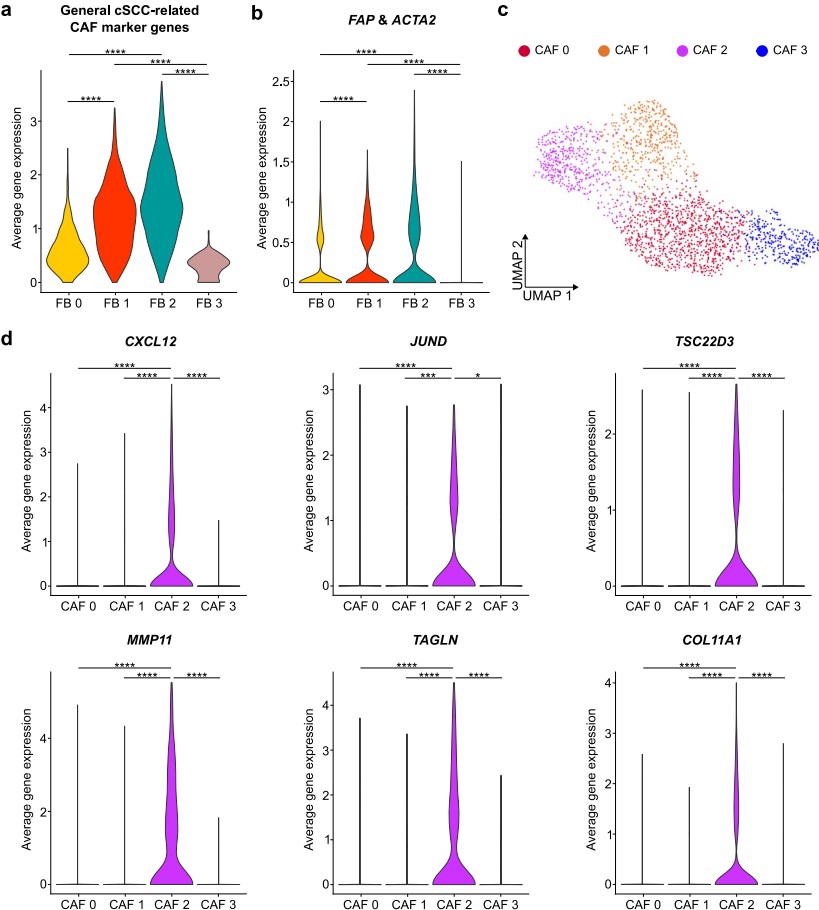

**Fig. 5 | Functional separation of BCC-related CAFs differs from cSCC. a** Violin plot showing the average gene expression of cSCC-related CAF marker genes in fibroblasts that were detected in the integrated analysis of nine BCC samples from Yerly et al. [46], and Guerrero-Juarez et al. [45]. All indicated comparisons: *p* value < 2.2e-16. **b** Violin plot showing the average gene expression of the known CAF marker genes *FAP* and *ACTA2* in fibroblasts from all BCC samples. All indicated comparisons: *p* value < 2.2e-16. **c** UMAP plot visualizing four actual CAF clusters. **d** Violin plots depicting the average gene expression of cSCC-related iCAF (*CXCL12*, *JUND*, *TSC22D3*) and myCAF (*MMP11*, *TAGLN*, *COL11A1*) marker genes in the detected CAF clusters from BCC. *CXCL12* CAF 0 vs. 2: *p* value < 2.2e-16, *CXCL12* CAF 1 vs. 2: *p* value = 2.78e-06, *CXCL12* CAF 2 vs. 3: *p* value < 2.2e-16, *JUND* CAF 0 vs. 2: *p* value =

8.37e-09, *JUND* CAF 1 vs. 2: *p* value = 0.0002782, *JUND* CAF 2 vs. 3: *p* value = 0.0115, *TSC22D3* CAF 0 vs. 2: *p* value < 2.2e-16, *TSC22D3* CAF 1 vs. 2: *p* value = 4.40e-12, *TSC22D3* CAF 2 vs. 3: *p* value = 1.00e-15, *MMP11* CAF 0 vs. 2: *p* value < 2.2e-16, *MMP11* CAF 1 vs. 2: *p* value < 2.2e-16, *MMP11* CAF 2 vs. 3: *p* value < 2.2e-16, *TAGLN* CAF 0 vs. 2: *p* value < 2.2e-16, *TAGLN* CAF 1 vs. 2: *p* value < 2.2e-16, *TAGLN* CAF 2 vs. 3: *p* value < 2.2e-16, *COL11A1* CAF 0 vs. 2: *p* value = 4.47e-15, *COL11A1* CAF 1 vs. 2: *p* value = 5.57e-15, *COL11A1* CAF 2 vs. 3: *p* value < 2.2e-16). For violin plots, statistical analyses were performed using a two-sided Wilcoxon Rank Sum test (*: *p* value < 0.05, ***: *p* value < 0.001, ****: *p* value < 0.0001). FB fibroblast, iCAF inflammatory cancer-associated fibroblast, myCAF myofibroblastic cancer-associated fibroblast, BCC basal cell carcinoma, cSCC cutaneous squamous cell carcinoma.

separation of the two types of CAFs in our multiplexed RNA FISH assays, as previously described in other cancer types[23,25]. In PDAC, for example, myCAFs were detected only closely surrounding cancer cells[23]. Also, in a murine breast cancer study, one CAF subpopulation was detected mostly surrounding blood vessels, and appeared to be involved in vascular development and angiogenesis[25]. The fact that the CAFs in cSCC are distributed throughout the TME could be due to differences in the distributions of the cell types that can give rise to CAFs. In this regard, while cSCC-derived CAFs arise from resident dermal fibroblasts[29], myCAFs in PDAC are known to develop from pancreatic stellate cells[23], and vascular CAFs in the abovementioned murine breast cancer model were believed to originate from specific perivascular cells[25].

On the other hand, it is important to point out that our scRNA-seq analysis suggested that the proportions of both CAF subtypes may dynamically change during tumorigenesis, with more immunoregulatory iCAFs in the in situ carcinomas and more myCAFs in the invasive cSCCs. This seems plausible, as ECM-remodeling capabilities may be most necessary for invasive tumor stages. The potential increase of the myCAFs upon cSCC progression is also supported by

our multiplexed RNA FISH experiments, and also by the higher expression of the mesenchymal gene signature towards the end of the continuum. A higher presence of iCAFs in early stages could not be confirmed with RNA FISH assays, although this could simply be explained by the few markers used in these experiments, and the fact that none of these markers are expressed in all cells of each particular subtype. Interestingly, our cell-cell communication analyses also suggested that immunoregulatory CAFs have a larger impact on the TME and especially on the immune system in early tumor stages, while ECM-remodeling CAFs appeared more impactful in later tumor stages, promoting matrix re-organization.

Multiple cellular pathways, such as the JAK/STAT3 and PI3K/AKT/mTOR signaling cascades are known to be involved in the CAF-dependent establishment of a pro-tumorigenic TME via interactions with cancer cells and nonmalignant cell types[54]. Interestingly, our analyses suggest that the activated pathways depend on the CAF subtype. In general, we observed CAFs as the key secreting cell type in the TME, with both subtypes promoting proliferation and invasion in cancer cells. However, our results also predict that these cells are able to activate distinct and specific signaling pathways in nonmalignant

cell types, supporting their functional separation into iCAFs and myCAFs. The iCAF-related CXCL12/CXCR4 pathway is involved in immunoregulatory functions, whereas myCAFs activate p38 MAPK signaling, leading to enhanced ECM remodeling. Moreover, myCAF-derived WNT5A also appears to be correlated with tissue invasion in KCs[45,55]. Together, these cellular interactions result in a tumor promoting microenvironment.

In conclusion, our study constitutes a thorough analysis of CAF development during cSCC tumorigenesis, thus providing important and comprehensive insights into the potential origins and specific functional characteristics of these key cells within the TME. In several cancers, CAFs have been described to be associated with poorer prognosis, therapy resistance, and disease recurrence[19]. In addition, several studies already investigated the prognostic value of general CAF biomarkers, such as FN1 and COL3A1[56]. A better understanding of the functional CAF heterogeneity along the cSCC continuum might therefore improve the accuracy of disease prognosis and aid the development of stage-dependent therapies. Also, the increased signaling activity observed for iCAFs in BD and for myCAFs in cSCC could provide interesting therapeutic targets for specific CAF populations in BD and cSCC. An increasing number of studies and clinical trials aim for the combined targeting of malignant cells (tumor "seeds") and CAFs as their fertilizing tumor "soil"[19]. For example, a phase II trial has shown the beneficial effects of combining an anti-PD-1 therapy with CXCR4 blockade to promote T cell tumor infiltration in PDAC[57]. Furthermore, depletion of ECM-related CAFs and pro-angiogenic endothelial cells in triple-negative breast cancer has also resulted in decreased cancer cell proliferation and survival benefits in murine models[58]. While further studies will be needed to determine CAF subpopulation-specific clinicopathological characteristics of cSCC, our data should lead to a functional investigation of the CAF subtypes involved in skin tumorigenesis, and their potential value as prognostic biomarkers and therapeutic targets.

## Methods

### Human skin samples

All skin specimens were obtained from remnant tissue, not required for diagnostic purposes, of patients undergoing routine surgery at the Department of Dermatology, University Hospital of Heidelberg, after written informed consent by the patient and as approved by the Ethics Committee of Heidelberg University (S-091/2011) in compliance with the current legislation and institutional guidelines. No participant compensation was provided. All histological diagnoses were reviewed by a dermatohistopathologist before inclusion in this study. All BD samples originated from extragenital, UVR-exposed body regions, and were classified following current guidelines[59] as in situ carcinomas displaying atypical and polymorphic keratinocytes throughout the epidermis, while maintaining the basement membrane.

For scRNA-seq, 3 mm punch biopsies were obtained from chronically UVR-exposed healthy skin (n = 3), extragenital BD (n = 3), and cSCC (n = 5) samples from male participants older than 47 years. Clinicopathological characteristics are summarized in Supplementary Table 1.

For RNA FISH analyses, FFPE skin samples of male patients (>65 years) diagnosed as UVR-exposed healthy skin (n = 3), AK (n = 3), BD (n = 3) and cSCC (n = 5) were retrieved from the tissue bank of the Department of Dermatology, University Hospital of Heidelberg, and cut into consecutive 4 μm sections at the tissue bank of National Center for Tumor Diseases Heidelberg (NCT), Germany. A hematoxylin and eosin (H&E) stained 4 μm section of each specimen was digitalized for microanatomical correlation by virtual microscopy. Clinicopathological characteristics are summarized in Supplementary Table 3.

### Sample preparation for scRNA-seq

Fresh human skin tissue was kept in MACS Tissue Storage Solution (Miltenyi Biotec, cat. no. 130-100-008) at most overnight. It was then enzymatically and mechanically dissociated using the Whole Skin Dissociation kit for human material (Miltenyi Biotec, cat. no. 130-101-540) and the Gentle MACS Dissociator device (Miltenyi Biotec), following the manufacturer's instructions. The resulting cell suspensions were filtered through 70 μm cell strainers (Falcon, cat. no. 10788201) and the Dead Cell Removal Kit (Miltenyi Biotec, cat. no. 130-090-101) was then used to remove apoptotic and dead cells. Subsequently, about 20,000 single cells per sample were used to prepare sequencing libraries with the Chromium Single Cell Reagent kit (v3.1 chemistry) from 10X Genomics (cat. no. 1000128), following the manufacturer's instructions. Library quality was checked with the Qubit dsDNA HS Assay kit (Life Technologies, cat. no. Q32851), and cDNA integrity was assessed with D1000 ScreenTapes (Agilent Technologies, cat. no. 5067-5582). Finally, pairwise sequencing (26 + 96 bp) was performed with a NovaSeq 6000 device (Illumina).

### scRNA-seq data analysis

Raw sequencing reads from all samples were processed with the CellRanger v4.0.0 software from 10X Genomics and then analyzed using the Seurat v4.0.5 package[60] in R v4.0.3 (R Core Team, 2020), including also reads from three UVR-protected healthy skin samples of older donors from Solé-Boldo et al. [27]. If not otherwise specified, for all functions the default parameters were used. To improve data quality, only cells expressing between 200 and 2000 unique genes were initially kept for subsequent analyses. In addition, cells with >5% mitochondrial reads were discarded, which resulted in a final dataset of 115,053 unique cells.

Following the standard integration protocol described in Seurat (to avoid batch effects), log-normalization of unique molecular identification (UMI) counts and identification of the 2000 most variable genes per sample were performed. Integration anchors were then calculated for the 14 datasets and integration was performed with the first 80 dimensions of canonical correlation analysis (CCA). Next, the integrated data was scaled and the dimensions of the principal component analysis (PCA) were calculated. A K-nearest neighbor (KNN) plot was constructed with the first 130 PCA dimensions and then used for the unsupervised clustering of the cells with a resolution of 0.9. For final visualization, the non-linear dimensional reduction UMAP was ultimately performed with the first 130 PCA dimensions.

The most differentially expressed genes between all cell clusters (and therefore considered as cluster marker genes) were identified using the FindAllMarkers function taking into consideration only genes that are detected in a minimum fraction of 25% of the cells (min.pct = 0.25). These detected marker genes were used for cell type annotation, together with markers found in the literature for all cell types typically present in human skin[27] (Supplementary Fig. 1a).

Cell cycle analyses were performed using the CellCycleScoring function in Seurat, which assigned a cell cycle score to each cell, based on the expression of G2/M and S phase marker genes from Tirosh et al. [29]. These scores were used to predict classifications for each cell in either G2/M, S, or G1 phase, assuming that cells expressing neither G2/M nor S phase marker genes are likely in G1 phase.

Cell proliferation analyses were performed calculating the average expression for each cell of a proliferation-related gene signature defined by Whitfield et al. [61], comprising *MKI67*, *MYBL2*, *BUB1*, *PLK1*, *CCNE1*, *CCND1* and *CCNB*.

Similarity scores for PDAC-related iCAF and myCAF gene signatures[23,24] were calculated using the AddModuleScore function in Seurat. It calculates the average expression of target gene signatures within different clusters/groups and subtracts the aggregated expression of randomly selected control genes present in the cells.

## CNV inference

To infer CNV events in keratinocytes of the scRNA-seq datasets, the R packages InferCNV v1.6[29,32] and CopyKAT v1.1[62] were used, following the provided tutorials. The algorithms compare gene expression intensities in cancer cells, and across all chromosomes, with healthy reference cells, after several filtering, normalization and adjustment steps. In our case, keratinocytes from all UVR-protected healthy skin samples were used as reference cells, and their gene expression was compared with that of keratinocytes from healthy chronically UVR-exposed skin, BD and cSCC samples. Supplementary Figure 2a shows the CNV events in a heatmap generated with InferCNV, while CopyKAT was used to classify BD- and cSCC-derived keratinocytes as either aneuploid or diploid (Supplementary Fig. 2b).

## GO analysis

All GO analyses were performed with the Gene Functional Annotation Tool from the DAVID Bioinformatics Database v6.8[63,64]. The most representative genes ($p$ value < 0.05 and fold change ≥1.5) expressed in each cell group were used as input. Then, the option GOTERM_BP_DIRECT was selected, and the top significant ($p$ value < 0.05) terms were visualized in bar plots, sorted according to their $p$ values.

## GSEA

All GSEA plots were generated with the GSEA software using gene sets from the Molecular Signatures Database (MSigDB) v2023.1.Hs[65,66].

## Trajectory inference

In order to analyze the progression from healthy fibroblasts towards CAFs, a gene expression-based trajectory inference was performed using the scRNA-seq data. First, for each sample type the respective scRNA-seq datasets were normalized, merged and batch corrected for the different patients using the Combat method[67]. Then, data from PH, BD and cSCC samples were combined and analyzed following the standard protocol of the R package Seurat v4.0.5[60], with default parameters if not otherwise specified. This included the detection of variable features, scaling the data and performing a PCA. The cells were then clustered using the first 20 PCA dimensions and a resolution of 0.1, before performing the non-linear dimensional reduction UMAP also with the first 20 PCA dimensions. Subsequently, cell types were assigned using the information obtained from the integrated Seurat object from Fig. 1a and Supplementary Fig. 1a. After isolating fibroblasts, the slingshot function from the R package Slingshot v1.8[68] was used to infer an unsupervised trajectory between healthy fibroblasts in UVR-protected skin and CAFs from BD and cSCC, based on the first two UMAP dimensions. In addition, the algorithm projected the cells onto a pseudotime axis visualizing the progression.

## Cell-cell communication analysis

To infer cell-cell communication patterns in BD and cSCC tumor samples, the R package CellChat v1.4[37] was used, following the provided tutorial. The algorithm makes use of the scRNA-seq gene expression data to model cell-cell interaction probabilities based on database knowledge on interacting signaling ligands, receptors and co-factors. The analyses were performed for BD and cSCC samples separately, using all cell types that presented enough input cells. In this regard, melanocytes, Schwann cells and erythrocytes were removed from the analyses due to their low numbers. Suggested signaling pathways involving iCAFs or myCAFs as key "sender" cells were subsequently analyzed in greater detail.

The R package LIANA v0.1.11[44] was further used to confirm the specificity of the defined interactions in cSCC, following the provided tutorial. This tool combines several interaction analysis methods and resources, and its output is based on their consensus. Also here, melanocytes, Schwann cells and erythrocytes were removed from the analyses due to their low numbers.

## Multiplexed RNA FISH analysis

To detect the expression of specific CAF-related genes in human skin samples throughout cSCC tumorigenesis, the RNAscope HiPlex12 Reagents v2 kit (Bio-Techne, cat. no. 324442) was used, and 11 probes were specifically designed: *APCDD1* (Bio-Techne, cat. no. 535851-T1), *POSTN* (Bio-Techne, cat. no. 409181-T2), *MMP11* (Bio-Techne, cat. no. 479741-T3), *COL3A1* (Bio-Techne, cat. no. 549431-T4), *APOE* (Bio-Techne, cat. no. 433091-T5), *IGF1* (Bio-Techne, cat. no. 313031-T6), *CCN5* (Bio-Techne, cat. no. 888111-T7), *C3* (Bio-Techne, cat. no. 430701-T8), *WNT5A* (Bio-Techne, cat. no. 604921-T10), *DIO2* (Bio-Techne, cat. no. 562211-T11) and *KRT14* (Bio-Techne, cat. no. 310191-T12). In addition, the following probe set was used to assess cellular interactions between CAFs and nonmalignant cell types: *CXCR4* (Bio-Techne, cat. no. 310511-T2), *CXCL12* (Bio-Techne, cat. no. 422991-T3), *COL3A1* (Bio-Techne, cat. no. 549431-T4), *IGF1* (Bio-Techne, cat. no. 313031-T6), *MCAM* (Bio-Techne, cat. no. 601731-T7), *C3* (Bio-Techne, cat. no. 430701-T8), *CD3D* (Bio-Techne, cat. no. 599391-T9), *WNT5A* (Bio-Techne, cat. no. 604921-T10), *VWF* (Bio-Techne, cat. no. 560461-T11) and *KRT14* (Bio-Techne, cat. no. 310191-T12). Prior to the experiments, all FFPE sections, which were obtained from male individuals with similar ages to those included in our scRNA-seq analysis (Supplementary Table 3), were stained with H&E to further confirm their diagnosis and delineate tumor regions. Assays were performed in multiple rounds, following the kit manufacturer's instructions.

Briefly, 4 μm FFPE sections of all samples were deparaffinized using xylene followed by 100% ethanol, and then subjected to a target retrieval step and protease treatment. Subsequently, all probes were hybridized with the tissue samples. After several signal amplification steps, fluorophores for the first four probes were also coupled and sections were counterstained with DAPI. For mounting, the ProLong Gold Antifade Mountant (ThermoFisher, cat. no. P36930) was used, and images were taken with a slide scanner microscope Slideview VS200 (Olympus) using its 40X objective. Coverslips were then removed and imaged fluorophores were cleaved. This procedure of fluorophore coupling, DAPI staining, mounting and imaging was repeated twice. All images were further processed using QuPath v0.3.2[69], the Fiji software v2.3.0[70] and the HiPlex Image Registration Software v2.0.1 from Bio-Techne.

To quantify CAFs in BD and cSCC sections, *COL3A1*+ cells were counted in multiple dermal regions (each dot in Figs. 3b and 3c represents a dermal region with around 100 *COL3A1*+ cells). Co-expression of *COL3A1* with either *C3*, *IGF1* or both, or with *MMP11*, *WNT5A* or both, was used to assess the amount of iCAFs and myCAFs, respectively. CAFs showing mixed co-staining for iCAF and myCAF marker genes were not considered during the quantification, since a reliable subtype classification was not possible. Furthermore, co-expression of iCAF/myCAF marker genes with the fibroblast subpopulation markers *APCDD1* and *DIO2* (secretory-papillary), *CCN5* (also known as *WISP2*, secretory-reticular), *APOE* (pro-inflammatory) and *POSTN* (mesenchymal) was quantified to analyze the potential origin of both CAF subgroups. Cells showing co-staining for *POSTN* and *APOE* were filtered out during quantification, since a clear cell type assignment was not possible.

To quantify cellular interaction between iCAFs/myCAFs and non-malignant cell types, T cells or vascular endothelial cells were counted in multiple dermal regions (each dot in Supplementary Figures 8f and g represents a region with around 100 T cells or vascular endothelial cells, respectively). Next, the amount of iCAFs (*COL3A1*+, *C3*+) and myCAFs (*COL3A1*+, *WNT5A*+) was determined in the direct surrounding.

## Statistics

Statistical analyses of scRNA-seq data were performed using CellRanger and the Seurat package in R. To compare the fibroblast proportions in the scRNA-seq datasets corresponding to healthy UVR-protected skin, healthy chronically UVR-exposed skin, BD, and cSCC samples, one-way ANOVA with pairwise comparisons was performed using the Holm-

Sidak test in SigmaPlot v14.0. Cell cycle ratios of mesenchymal fibroblasts in all entities were also quantified with one-way ANOVA. Proportions of iCAFs and myCAFs in the BD and cSCC samples, the correlation of both CAF subtypes with the different fibroblast subpopulations, as well as iCAFs and myCAF ratios at each cell cycle phase were analyzed with two-sided unpaired and paired t-tests. RNA FISH quantifications were also analyzed in the latter manner.

## Reporting summary

Further information on research design is available in the Nature Portfolio Reporting Summary linked to this article.

## Data availability

The scRNA-seq data generated in this study have been deposited in the Gene Expression Omnibus (GEO) database under accession code GSE218170. The scRNA-seq publicly available data used in this study are available in the GEO database under accession codes GSE130973[27], GSE181907[46] and GSE141526[45]. The GO and GSEA publicly available data used in this study are available in the DAVID Bioinformatics Database v6.8[63,64] and Molecular Signatures Database v2023.1.Hs[65,66], respectively. The remaining data are available within the Article, Supplementary Information or Source Data file. Source data are provided with this paper.

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

## Acknowledgements

We gratefully acknowledge the help of Dr. Günter Raddatz (Division of Epigenetics, DKFZ) with the alignment of the raw scRNA-seq reads, as well as the help of Dr. med. Enrico Streit and Dr. med. David Deltgen for dermatosurgery, and the clinical staff of the Department of Dermatology at Heidelberg University Hospital. We also thank Fabio Tabone (Tissue Bank of the National Center for Tumor Diseases (NCT), Heidelberg, Germany) for technical support, Dr. Mireia Berdiel-Acer for the fruitful scientific discussions, and the DKFZ Single-Cell Open Lab (scOpenLab) for assistance with the scRNA-seq and RNA FISH experiments.

## Author contributions

S.S., L.S.-B., A.S.L., M.R.-P. and F.L. analyzed the data. S.S. performed scRNA-seq and RNA FISH assays. A.S.L., J.H., C.L.-P. and A.B. provided clinical samples, technical assistance and medical expertise. M.R.-P. and F.L. conceived the study. S.S., M.R.-P. and F.L. wrote the paper with input from other authors. All authors read and approved the final manuscript.

## Funding

## Competing interests

The authors declare no competing interests.
