## [Peer Review File · Nature Communications]

Reviewers' Comments:

Reviewer #1:

Remarks to the Author:

This article by Schutz and colleagues characterizes the transcriptional profile of fibroblasts in skin, precancerous lesions and cSCC. This is important to understand as the tumor stromal microenvironment likely plays an important role in tumorigenesis.

The following issues should be addressed:

1. Abstract, second sentence, AK can progress to SCCIS and SCCIS can progress to cSCC, the probability of each progressing is low so better to say AK/SCCIS may progress.
2. Bowen's disease is typically used for SCCIS with specific features and often found on sun-protected skin, probably more accurate to replace this term with SCCIS unless you only analyzed classical BD specimens.
3. Intro, line 54, The epidermis is stratified squamous epithelium, the term 'palisade' is not typically used to describe the epidermis.
4. Intro, the authors may want to incorporate some discussion of UVA since it does cause damage to the epidermis and the superficial dermis where the fibroblasts that become CAFs are found.
5. Intro, line 61 may be better to say 'leading to DNA mutations', not sure what an epigenetic mutation means in the context.
6. Intro, line 67, lifetime risk is being conflated with annual incidence. The numbers are a bit off. Annually in the US there are probably at least 700K cases of cSCC; the number of patients who have had cSCC is in the tens of millions. Logically, you can argue that the actual incidence is close to 1M cases per year since not all cSCCs are reported.
7. Melanoma deaths are around 7-8K per year, cSCC deaths are estimated to be 3-4K per year. Please clarify your sources.
8. Fig 1C. What is the total cell number for each category. Also, what is the average cell number per specimen in the PH, EH, BD and cSCC categories? If the number of fibroblasts per cSCC specimen is much higher, than what is the significance of these proportions?
9. Line 159, again BD is carcinoma in situ and not malignant. The lesion is considered pre-cancerous. Wordage used is a bit confusing.
10. Fig 3. Please indicate the junction between SCC dermis and unremarkable dermis and BD dermis and unremarkable dermis in your samples. Your human biopsies likely have this junction. It is important to show the distribution and number of iCAF and myCAF in adjacent dermis underlying unremarkable skin.
11. The labeling of the x axis in the violin plots is difficult to interpret. Can the authors compare the BCC fibroblasts with the cSCC fibroblasts directly in a graph to show how they differ?

Reviewer #2:

Remarks to the Author:

Schütz et al. presented a single cell RNA-seq study where they integrated three healthy sun-exposed skin samples, three BD and five invasive cSCC, along with previously published three healthy sun-protected (unexposed) skin samples. In particular, the authors further focused on the fibroblast cells, especially cancer associated fibroblasts (CAFs), and showed inflammatory CAFs (iCAF) and myofibroblastic CAFs (myCAF) and their associated potential "origin". Although the

manuscript itself reads well, there were a lot of places that require further clarification and analysis to make the results and key findings robust. Some places were contradicting each other, not supporting their claims in the paper. The specific comments are as follows.

1. Although the focus is fibroblasts in this paper, authors need to present the overview of all the cells detected and compare with what has been published, e.g., Ji et al. Cell 2020, Reynolds et al., Science 2021, others, in terms of proportions of major cells identified, to show their dataset is similar or different from published data.

More importantly, the authors need to further characterise their keratinocyte populations, and show the profiles of keratinocytes from BD and cSCC are indeed cancer derived keratinocytes (e.g., using Ji et al. Cell 2020 as a reference). All the keratinocytes of BD and cSCC were clustered together with keratinocytes from health tissue, but were they present in separate sub-clusters of keratinocytes as one should expect? Are signatures of cancer derived keratinocytes absent in normal keratinocytes? This QC step is needed to assure the quality of the data.

2. RE CNV profiling. Since these samples are from different patients, it is surprising to see they shared so many "recurrent" CNAs across samples, e.g., chr1q gain, chr6p gain, chr14q gain, and some chromosome arm losses. However, these were not common recurrent CNAs identified previously by SNP array or whole exome sequencing (e.g., Inman et al Nature Comms 2018). It is likely there were many artefacts in the CNVs identified.

3. P7 Line 152, is ANOVA an appropriate test here if you only compared cSCC tumours against healthy, UVR-protected skin samples?

4. RE Cell cycle / proliferation analysis, e.g., Supplementary Figure 2C. How did authors assign the cells to each cell cycle phases? it was unclear how this was done. The authors should use the cell cycle and proliferation gene sets, and calculate the gene set activity/score using AddModuleScore for each cell for examples, and show in boxplot or violinplot or feature plot to compare the cell cycle/proliferation activities amongst four fibroblast types and other subtypes.

5. The authors suggested the 'origin' of CAFs by simply looking at the expression correlation of signature gene sets with the four fibroblast types identified from normal skin. However, this 'origin' should be used with care, as one cannot suggest the origins without proper cell tracing in vivo experiments. Simply looking at the expression of signature genes was just too weak.

6. Can the authors perform trajectory analysis to confirm the evolution of CAFs in BD and cSCC patients, with the normal skin as a reference/comparator?

7. The authors identified four fibroblast subpopulations in the integrated analysis of normal skin, BD and cSCC samples, with all samples contributing to the four subtypes. They said these four fibroblast subtypes were previously identified and defined in the healthy, UVR-protected human skin of their previous study. They further focused on the fibroblasts of BD and cSCC samples, and identified iCAF and myCAF two CAF populations out of all fibroblasts of BD and cSCC. Does this imply that CAF signatures were also present in normal skin samples? The analysis and approaches used here seemed to contradict each other.

8. Figure 3B showed there were many more myCAFs than iCAFs in both BD and cSCC based on their RNA FISH analysis. The myCAF proportions were also very similar between BD and cSCC. This did NOT support their claim that myCAFs increased with disease progression, with more myCAFs in cSCC.

9. Figure 3D showed POSTN positive cells (mesenchymal fibroblasts) were higher in BD than in cSCC. But this is contradicting to Figure 1C where mesenchymal fibroblasts were highest in cSCC.

10. n=3 AKs were used in the RNA FISH experiment. This is not enough and weak to support their claims that the two CAFs were absent from AKs. In fact, Supp Figure 4b and 4c, AK sample multiplex FISH is positive for iCAFs and myCAFs marker, which suggest that these two-tumour supporting CAF subpopulation is not absent in all AK, can author explain this?

It is unfair to compare the CAF profile of BD and cSCC derived from scRNA-seq with CAF signature of AKs derived from RNA FISH. The authors should include scRNA-seq data of AKs to confirm the preservation of four subtypes of fibroblasts and absence of iCAFs and myCAFs in AK.

11. Furthermore, to address the issue of small sample size of each clinical group, authors should perform bulk tissue RNA-seq deconvolution analysis using their identified single cell signatures using rich published RNA-seq data sets, to confirm their findings of fibroblast and CAF abundance and signatures.

12. Can author provide further proof for ADM-CALCRL and CXCL12-CXCR4 interaction in iCAFs, as well as ANGPTL2-ITGA5 and WNT5A-ITG5A interaction in myCAFs? Without any evidence, it is very difficult to believe the robustness of the results. Can authors at least try a couple more cell-cell interaction analysis methods, and could these results be replicated?

13. The final section where CAF signature were investigated and compared in BCC was weak. The authors did not specify clearly what methods and criteria they used to assign cSCC CAFs onto BCC cells. Based on Supplementary Figure 6D, one could say FB3 resembles iCAF the most, and FB2 resembled myCAF the most. Also, FB0 also had good level of CAF expression, and similar expression of FAP and ACTA2 to FB1 and FB2 (see Figure 5A&B). so it is unclear why FB0 was exclude in the analysis.

Furthermore, what was the reason/rationale why the authors wanted to investigate cSCC CAF signatures in BCC? Also 3998 fibroblast cells in BCC compared to 27382 fibroblast cells in cSCC was too difficult to derive a valid conclusion.

14. There seemed a lack of a key message of clinical importance of their findings. Can authors comment more on the clinical and functional implications?

A few minor comments,

1. The sample size is very small especially for healthy controls and BD samples. All male samples could lead to gender biased results which can't be translated for general population. Also, author should include information about patient's immune status, if its IS or IC? Further analysis of scRNA-seq data based on IC or IS samples could have impact on iCAFs and myCAFs population in cSCC

2. Can author provide more information how the cluster markers were identified? Parameters used to calculate differentially expressed markers. In fact, the bioinformatics analysis was very sparse in the methods.

3. Author performed strict quality controls and followed stringent parameters to discard unhealthy or outlier cells. However, author should apply doublet removal algorithm to all samples and confirm that the only singlet cells were used for downstream clustering and analysis. The cell numbers among the 14 samples are quite drastic and its highly likely that samples with very high cell number could have significant number of doublets. Author used CCA Seurat method to integrate different samples.

4. Can author perform GSEA analysis on iCAFs and myCAFs population with respective marker to further corroborate the population identity?

5. There are a few labelling errors of figure panels throughout the paper.

Authors uploaded all unprocessed and processed Seurat objects along with all raw data. This is great!

Reviewer #3:

Remarks to the Author:

The study includes identification of two main scRNA seq-based CAF subsets in cSCC with some similarities to iCAFs and myCAFs, and to earlier described skin fibroblast subsets (Fig. 2). These subsets are validated by multiplexed FISH using a handful of probes in cSCC and BD samples (Fig. 3). Bioinformatics approaches are used to deduce possible functions of the two CAF subsets (Fig. 4). Finally, analyses of publicly available datasets are done, leading to the conclusion that CAF

subsets in BCC are distinct from the two main cSCC CAF subsets (Fig. 5).

The study is overall well-structured and conclusions are in general supported by data.

In summary, the study presents a set of potentially important data regarding functional properties and clinical significance of tentative cSCC CAF subsets. However, as of now the study lacks the novelty and significance associated with studies published in Nat Comm.

Some suggestions regarding expansion and development of findings are presented below.

1. The similarities of the two main subsets to iCAFs and myCAFs could be substantiated by looking at their similarity scores to published iCAF and myCAF signatures, like the ones in e.g. Elyada, Cancer Discovery, 2019.
2. Figure 3a, which is used for in situ validation requires staining at the same time with all four markers to allow the conclusion that two subsets (C3+/IGF1+/MMP11-/Wnt5A- and C3-/IGF1/MMP11+/Wnt5A+) have been "in-situ-validated".
3. The in situ analyses (Fig. 3) should be expanded to include some information on differential localization of the two main subsets in relationship to e.g. malignant cells, T-cells, macrophages and vessels.
4. Some data on how iCAF/myCAF abundance in different cases is associated with clinical characteristics, including survival, would add to the study.
5. The data from the bioinformatics analyses (Fig. 4) on cell-cell communications suggests a series of in situ analyses where the proposed inter-cellular communications could be validated.
6. The preliminary indications that the two CAF subsets show different similarities to the four fibroblasts subsets of normal skin (Fig. 1) is an interesting finding that merits further attention.

Reviewer #1:

1. Abstract, second sentence, AK can progress to SCCIS and SCCIS can progress to cSCC, the probability of each progressing is low so better to say AK/SCCIS may progress.

The sentence has been corrected in the abstract, as suggested. Accordingly, we have now also modified a sentence in the Introduction section.

2. Bowen's disease is typically used for SCCIS with specific features and often found on sun-protected skin, probably more accurate to replace this term with SCCIS unless you only analyzed classical BD specimens.

This has now been clarified in the Methods section of the manuscript. The Bowen's disease samples that were included in our work were diagnosed by board-certified dermatohistopathologists following the guidelines described in Wolff *et al.*, 2011 (reference 59 of the manuscript). More specifically, they were all classified as extragenital *in situ* carcinomas originating from UVR-exposed regions, and distinguished from AK by the presence of atypical and polymorphic keratinocytes throughout all layers of the epidermis, while keeping intact the basement membranes.

3. Intro, line 54, The epidermis is stratified squamous epithelium, the term 'palisade' is not typically used to describe the epidermis.

This has been corrected in the text, as suggested. We have removed the term 'palisade' and the epidermis is now described as a stratified squamous epithelium.

4. Intro, the authors may want to incorporate some discussion of UVA since it does cause damage to the epidermis and the superficial dermis where the fibroblasts that become CAFS are found.

The Introduction section of the manuscript now includes some sentences that highlight the role of UVA radiation in skin cancer development.

5. Intro, line 61 may be better to say 'leading to DNA mutations', not sure what an epigenetic mutation means in the context.

We have changed the sentence for clarity. The manuscript now includes '...leading to genetic alterations'.

6. Intro, line 67, lifetime risk is being conflated with annual incidence. The numbers are a bit off. Annually in the US there are probably at least 700K cases of cSCC; the number of patients who have had cSCC is in the tens of millions. Logically,

you can argue that the actual incidence is close to 1M cases per year since not all cSCCs are reported.

The Introduction section of the manuscript now includes the corrected number of annual cSCC cases in the US. We have also revised the comparison with respect to melanoma cases.

7. Melanoma deaths are around 7-8K per year, cSCC deaths are estimated to be 3-4K per year. Please clarify your sources.

This has been corrected according to the new reference 13 of the manuscript.

8. Fig 1C. What is the total cell number for each category. Also, what is the average cell number per specimen in the PH, EH, BD and cSCC categories? If the number of fibroblasts per cSCC specimen is much higher, than what is the significance of these proportions?

All requested fibroblast numbers are now provided in the new Supplementary Table 4. On average, our scRNA-seq analysis included 1,327 fibroblasts per PH specimen, 4,546 fibroblasts per EH specimen, 1,269 fibroblasts per BD specimen, and 1,191 fibroblasts per cSCC specimen. The fact that some entities (or individual specimens) contribute a greater total number of cells to the analysis than others depends on technical issues, like tissue dissociation. However, this affects all subpopulations and does not introduce a bias into the proportions of the different cell types.

9. Line 159, again BD is carcinoma in situ and not malignant. The lesion is considered pre-cancerous. Wordage used is a bit confusing.

This has been corrected. Bowen's disease is no longer described as malignant throughout the manuscript.

10. Fig 3. Please indicate the junction between SCC dermis and unremarkable dermis and BD dermis and unremarkable dermis in your samples. Your human biopsies likely have this junction. It is important to show the distribution and number of iCAF and myCAF in adjacent dermis underlying unremarkable skin.

All RNA FISH microscopy images of BD and cSCC samples included in the original manuscript showed centered regions of the lesions. None of the BD tissue sections used for these assays contained clear areas of unremarkable skin, so unfortunately it is not possible to show the requested junctions for this entity. For two of the cSCC samples, non-tumoral skin was located at the edges of the cSCC tissue sections and could thus be directly compared. This is now included in our new Supplementary Figures 5c and 5d. The results show the absence of CAFs in the unremarkable skin regions of these two cSCC samples, as we had originally observed in healthy skin sections (see Figure 3a), which provides an important additional control.

11. The labeling of the x axis in the violin plots is difficult to interpret. Can the authors compare the BCC fibroblasts with the cSCC fibroblasts directly in a graph to show how they differ?

We have now added a direct comparison between cSCC- and BCC-related fibroblasts in the new Supplementary Figure 8c, with the violin plots showing the average gene expression of the indicated CAF, iCAF and myCAF signatures. We consistently observed a lower expression of all signatures in BCC-related fibroblasts compared to cSCC-related CAFs, which further substantiates our conclusions.

Reviewer #2:

1. Although the focus is fibroblasts in this paper, authors need to present the overview of all the cells detected and compare with what has been published, e.g., Ji et al. Cell 2020, Reynolds et al., Science 2021, others, in terms of proportions of major cells identified, to show their dataset is similar or different from published data.

More importantly, the authors need to further characterise their keratinocyte populations, and show the profiles of keratinocytes from BD and cSCC are indeed cancer derived keratinocytes (e.g., using Ji et al. Cell 2020 as a reference). All the keratinocytes of BD and cSCC were clustered together with keratinocytes from health tissue, but were they present in separate sub-clusters of keratinocytes as one should expect? Are signatures of cancer derived keratinocytes absent in normal keratinocytes? This QC step is needed to assure the quality of the data.

An overview of the cells detected is shown in Supplementary Figure 1.

Regarding the comparison of these results with those of other scRNA-seq studies, we consider the datasets from Ji *et al.*, 2020 (reference 30 of the manuscript) and Reynolds *et al.*, 2021 (reference 1 of this point-by-point response) as unsuitable, given the drastic differences in sample preparation. In both studies FACS sorting was performed with the dissociated tissue, and in Reynolds *et al.*, 2021 (reference 1 of this point-by-point response), epidermis and dermis were even processed separately. Therefore, to address this point we compared our data with those generated by Tabib *et al.*, 2018 (reference 31 of the manuscript), where scRNA-seq was performed on healthy human skin samples following a workflow comparable to our study. Overall, the proportions of the different cell types were similar, with minor differences that can be explained by the different biopsy collection sites. The results of this comparative analysis have been included in the manuscript as the new Supplementary Figure 1d.

Additionally, Gene Set Enrichment (GSEA) and Gene Ontology (GO) analyses confirmed the tumor features of the BD- and cSCC-derived keratinocytes, as they displayed higher expression of JAK/STAT signaling-related genes and genes involved in DNA repair, as well as a strong enrichment for functions such as angiogenesis, transcription (known to be elevated in tumors), aerobic

respiration, reaction to oxidative stress, and the activation of immune responses. Moreover, BD and cSCC keratinocytes showed an enhanced expression of proliferation-related genes that are important for the G2/M cell cycle phase, as defined by Tirosh *et al.*, 2016 (reference 29 of the manuscript). All these results are now shown in our new Supplementary Figures 2c to 2f. To further confirm the malignant status of the keratinocytes in our cSCC samples, we also included a new violin plot (new Supplementary Figure 2g) showing increased expression of the cSCC-derived keratinocyte gene signature previously defined by Ji *et al.*, 2020 (reference 30 of the manuscript).

The combined clustering of all keratinocytes is a consequence of the integration of the data from the different samples that is needed to remove batch effects. Simple merging of all datasets, without integration, results in specific keratinocyte clusters according to the different sample types and patients. This provides further confirmation for their identity as cancer-derived keratinocytes and is now shown in our new Supplementary Figures 2h and 2i.

2. RE CNV profiling. Since these samples are from different patients, it is surprising to see they shared so many "recurrent" CNAs across samples, e.g., chr1q gain, chr6p gain, chr14q gain, and some chromosome arm losses. However, these were not common recurrent CNAs identified previously by SNP array or whole exome sequencing (e.g., Inman *et al.* Nature Comms 2018). It is likely there were many artefacts in the CNVs identified.

CNV analyses were originally done to confirm the malignant status of keratinocytes from BD and cSCC samples. This is a standard approach in tumor scRNA-seq studies (references 29 & 32 of the manuscript), but not intended to accurately analyze CNVs in these cases.

Most of the tumors analyzed by Inman *et al.*, 2018 (reference 2 of this point-by-point response) were derived from immunosuppressed patients and only three samples (WD22, PD02, PD06) showed similar characteristics to the cSCC tumors included in our work (i.e., samples from old male immunocompetent patients and sun-exposed body areas). Interestingly, some of the recurrent CNVs in our cSCC analysis can also be observed in these three tumors, such as the gain of chr1q or the loss of chr9p, and most of the major CNV events mentioned in the main text of Inman *et al.* can also be detected in several of our cSCC samples, such as the gain of chr11q and chr3q, or the loss of chr3p and chr5q. Finally, some of the cSCC-related recurrent CNV events seen in our analysis also appear frequently in other studies (Salgado *et al.*, 2010; Sekulic *et al.*, 2010; Hameetman *et al.*, 2013; Purdie *et al.*, 2009; references 3-6 of this point-by-point response).

In addition, an independent R package (CopyKAT, Gao *et al.*, 2021; reference 62 of the manuscript) identified a similar pattern of recurrent CNV events in our samples, such as the gain of chr1q and chr14q, or the loss of chr9p. CopyKAT also predicted that, on average, 92% of all cSCC-derived keratinocytes included in our study could be considered aneuploid, further confirming their malignant status. These data have been included in the new Supplementary Figure 2b, and are now also explained in the Results section of the manuscript.

3. P7 Line 152, is ANOVA an appropriate test here if you only compared cSCC tumours against healthy, UVR-protected skin samples?

This statistical analysis was performed as follows: firstly, four different ANOVA tests were applied to compare the proportions of each fibroblast subpopulation between the four distinct sample types, UVR-protected healthy skin, UVR-exposed healthy skin, BD and cSCC. The presence of four groups (the sample types) in each of the four comparisons justifies the use of ANOVA. As this analysis yielded a significant change for the mesenchymal fibroblasts (p -value <0.05), additional pairwise comparisons were performed using the Holm-Sidak test to reveal the specific sample types between which the significant change was taking place. This showed that the significant difference occurred between the mesenchymal fibroblasts from UVR-protected healthy and cSCC samples (p -value <0.05). This has now been clarified in the Results section of the manuscript, as well as in the corresponding figure legend.

4. RE Cell cycle / proliferation analysis, e.g., Supplementary Figure 2C. How did authors assign the cells to each cell cycle phases? it was unclear how this was done. The authors should use the cell cycle and proliferation gene sets, and calculate the gene set activity/score using AddModuleScore for each cell for examples, and show in boxplot or violinplot or feature plot to compare the cell cycle/proliferation activities amongst four fibroblast types and other subtypes.

For the cell cycle analyses, which is now shown in Supplementary Figures 3d (previously 2c) and 4h (previously 3c), we used the CellCycleScoring function of the Seurat R package. With this function we first calculated cell cycle scores for each cell based on the expression of 43 and 54 S and G2/M phase marker genes, respectively, which were obtained from Tirosh *et al.*, 2016 (reference 29 of the manuscript). According to these scores, each cell was assigned to a cell cycle phase, S, G2/M or G1, assuming that cells expressing neither G2/M nor S phase marker genes were likely to be in G1. This has now been clarified in the Methods section of the manuscript, and violin plots displaying the expression levels of the gene signatures corresponding to the S and G2/M phases have now also been included as new Supplementary Figures 3c and 4g.

These new violin plots complement those that were already included in Supplementary Figures 3e (previously 2c) and 4i (previously 3c), which showed the average expression for each cell of a gene signature related to proliferation including *MKI67*, *MYBL2*, *BUB1*, *PLK1*, *CCNE1*, *CCND1* and *CCNB*, described by Whitfield *et al.*, 2006 (reference 61 of the manuscript). This is also now better explained in the Methods section of the manuscript.

5. The authors suggested the 'origin' of CAFs by simply looking at the expression correlation of signature gene sets with the four fibroblast types identified from normal skin. However, this 'origin' should be used with care, as one cannot suggest the origins without proper cell tracing in vivo experiments. Simply looking at the expression of signature genes was just too weak.

We agree with the reviewer and have now mentioned in the Discussion that *in vivo* lineage tracing would be required to adequately verify the origin/s of the defined CAFs. We also included an additional analysis based on pairwise comparisons with the most representative genes of iCAFs or myCAFs, and the four subtypes of dermal fibroblasts. Notably, the largest overlaps were detected between pro-inflammatory fibroblasts and iCAFs (15 genes), and between mesenchymal fibroblasts and myCAFs (22 genes). These results are shown in our new Supplementary Figure 4k and further support our claims about the CAF origins.

6. Can the authors perform trajectory analysis to confirm the evolution of CAFs in BD and cSCC patients, with the normal skin as a reference/comparator?

Trajectory analyses were performed and included in the new Supplementary Figure 4a. They show the development and evolution of CAFs in BD and cSCC patients from normal skin.

7. The authors identified four fibroblast subpopulations in the integrated analysis of normal skin, BD and cSCC samples, with all samples contributing to the four subtypes. They said these four fibroblast subtypes were previously identified and defined in the healthy, UVR-protected human skin of their previous study. They further focused on the fibroblasts of BD and cSCC samples, and identified iCAF and myCAF two CAF populations out of all fibroblasts of BD and cSCC. Does this imply that CAF signatures were also present in normal skin samples? The analysis and approaches used here seemed to contradict each other.

Indeed, fibroblasts from all analyzed entities showed gene expression patterns characteristic of one of the four previously defined subpopulations of dermal fibroblasts. In addition, fibroblasts specifically in Bowen's disease and cSCC samples also showed a CAF-related gene expression pattern that allowed us to classify them as iCAFs or myCAFs. Both observations together suggested that CAFs arise from pre-existing dermal fibroblasts during cSCC development.

Importantly, however, cSCC-related CAF gene expression patterns were not found to be enriched in fibroblasts from healthy skin. This result is now shown in our new Supplementary Figure 4j and the point is clarified in the text.

8. Figure 3B showed there were many more myCAFs than iCAFs in both BD and cSCC based on their RNA FISH analysis. The myCAF proportions were also very similar between BD and cSCC. This did NOT support their claim that myCAFs increased with disease progression, with more myCAFs in cSCC.

While both scRNA-seq and RNA FISH showed the same trend, neither approach provided statistically significant results. We have therefore toned down this aspect throughout the manuscript.

9. Figure 3D showed *POSTN* positive cells (mesenchymal fibroblasts) were higher in BD than in cSCC. But this is contradicting to Figure 1C where mesenchymal fibroblasts were highest in cSCC.

The apparent contradiction can be explained and resolved by the methodologies used: In Figure 1c, mesenchymal fibroblasts are identified based on the expression level of a gene signature comprised of several genes (*ASPN*, *POSTN*, *GPC3*, *TNN* and *SFRP1*). In contrast, the RNA FISH analysis in Figure 3d is only based on *POSTN* expression. In addition, cells showing a co-staining for *POSTN* and *APOE* were filtered out during signal quantification since a clear cell type assignment was not possible. This is now clarified in the Methods section of the manuscript.

10. n=3 AKs were used in the RNA FISH experiment. This is not enough and weak to support their claims that the two CAFs were absent from AKs. In fact, Supp Figure 4b and 4c, AK sample multiplex FISH is positive for iCAFs and myCAFs marker, which suggest that these two-tumour supporting CAF subpopulation is not absent in all AK, can author explain this?

It is unfair to compare the CAF profile of BD and cSCC derived from scRNA-seq with CAF signature of AKs derived from RNA FISH. The authors should include scRNA-seq data of AKs to confirm the preservation of four subtypes of fibroblasts and absence of iCAFs and myCAFs in AK.

Due to the small size of AK samples, all shaved material from patients is usually used for pathology, which greatly limits the availability of AK samples for our analyses. However, AK regions were also present on sections from two of the analyzed cSCC samples and did not reveal any RNA FISH staining for the general CAF marker *COL3A1*. These results provide further support for the notion that CAFs are not yet activated in AK, and are now shown in the new Supplementary Figure 6b.

11. Furthermore, to address the issue of small sample size of each clinical group, authors should perform bulk tissue RNA-seq deconvolution analysis using their identified single cell signatures using rich published RNA-seq data sets, to confirm their findings of fibroblast and CAF abundance and signatures.

Following the reviewer's suggestion, we set out to apply the CIBERSORTx algorithm (Newman *et al.*, 2019 and Steen *et al.*, 2020; references 7 & 8 of this point-by-point response) to infer the presence of the different healthy fibroblast and CAF subpopulations in several published bulk RNA sequencing datasets. Unfortunately, upon checking the materials and methods of the published papers that included the most promising datasets, we realized that only in Das Mahapatra *et al.*, 2020 (reference 9 of this point-by-point response) had the samples been collected in a comparable manner to those in our study. Further analysis of the samples in this work, however, also revealed that most of them were from immunocompromised or female patients, and/or had been taken from regions considered to be sun-protected. We, therefore, decided that it did not make sense to apply these algorithms with the currently available datasets.

More importantly, however, our manuscript includes multiplexed RNA FISH experiments that validate our scRNA-seq analyses on additional, independent tumor sections (see Figure 3 and Supplementary Figures 5 and 6).

12. Can author provide further proof for ADM-CALCRL and CXCL12-CXCR4 interaction in iCAFs, as well as ANGPTL2-ITGA5 and WNT5A-ITGA5 interaction in myCAFs? Without any evidence, it is very difficult to believe the robustness of the results. Can authors at least try a couple more cell-cell interaction analysis methods, and could these results be replicated?

To provide further proof for the specificity of the predicted iCAF and myCAF interactions with different non-malignant cell types in cSCC tumors, we have now also used the R package LIANA (Dimitrov *et al.*, 2022; reference 44 of the manuscript). This package comprises several interaction analysis methods and resources, thus providing observations based on their consensus. Our results obtained with this package suggested similar communication patterns between both CAF subtypes and keratinocytes through collagen and fibronectin secretion, iCAF specificity for ADM-CALCRL and CXCL12-CXCR4 interactions, and myCAF specificity for ANGPTL2-ITGA5+ITGB1 and WNT5A-MCAM interactions. These data have now been included in the new Supplementary Figure 7e.

In addition, the manuscript now includes multiplexed RNA FISH experiments that validate iCAF- and myCAF-related CXCL12-CXCR4 and WNT5A-MCAM interactions, respectively. See our new Figures 4d and 4e.

13. The final section where CAF signature were investigated and compared in BCC was weak. The authors did not specify clearly what methods and criteria they used to assign cSCC CAFs onto BCC cells. Based on Supplementary Figure 6D, one could say FB3 resembles iCAF the most, and FB2 resembled myCAF the most. Also, FB0 also had good level of CAF expression, and similar expression of FAP and ACTA2 to FB1 and FB2 (see Figure 5A&B). so it is unclear why FB0 was exclude in the analysis.

Furthermore, what was the reason/rationale why the authors wanted to investigate cSCC CAF signatures in BCC? Also 3998 fibroblast cells in BCC compared to 27382 fibroblast cells in cSCC was too difficult to derive a valid conclusion.

The aim of including a CAF analysis of BCC was to investigate whether the observed cSCC-related iCAFs and myCAFs could have similar counterparts in the other major type of keratinocyte cancer. To do so, we actually compared 3,998 BCC-derived fibroblasts from two publicly available scRNA-seq studies (Guerrero-Juarez *et al.*, 2022 and Yerly *et al.*, 2022; references 45 & 46 of the manuscript) with the 5,956 cSCC-derived fibroblasts from our work (please note that 27,382 is the total number of fibroblasts in all our samples).

After the unsupervised clustering of the integrated BCC datasets using Seurat, we analyzed whether the gene expression signatures of the cSCC CAFs were also present in the identified BCC fibroblasts. The results showed a lower

expression level of the cSCC-derived CAF signatures (both general, iCAF- and myCAF-specific) in BCC fibroblasts (see our new Supplementary Figure 8c).

To focus on the question whether BCC presents CAFs that are similar to those observed in cSCC, we further analyzed the two subpopulations (FB1 and FB2) that more clearly expressed both the cSCC-related CAF signature and the established CAF markers *FAP* and *ACTA2* (as seen in Figures 5a and 5b). Even these two subpopulations did not reveal cSCC iCAFs and myCAFs (Supplementary Figure 8e). Taken together, these results reinforce our conclusion that cSCC CAFs differ from BCC CAFs. This has now been clarified in the Results section.

14. There seemed a lack of a key message of clinical importance of their findings. Can authors comment more on the clinical and functional implications?

The new version of the manuscript now includes a paragraph in the Discussion section explaining the clinical significance of our work.

Minor comments:

15. The sample size is very small especially for healthy controls and BD samples. All male samples could lead to gender biased results which can't be translated for general population. Also, author should include information about patient's immune status, if its IS or IC? Further analysis of scRNA-seq data based on IC or IS samples could have impact on iCAFs and myCAFs population in cSCC

All samples were taken from male immunocompetent patients, as this group constitutes the classical majority of patients. Additional information about the samples has now been included in Supplementary Tables 1 and 6.

We would also like to emphasize that we compensated for the relatively small sample size of our scRNA-seq analyses by orthogonally validating the most important observations using multiplexed RNA FISH on additional independent samples.

16. Can author provide more information how the cluster markers were identified? Parameters used to calculate differentially expressed markers. In fact, the bioinformatics analysis was very sparse in the methods.

The most differentially expressed genes between the different clusters (and therefore considered as cluster marker genes) were identified using the FindAllMarkers function of Seurat. For this, the default parameters were used, except for min.pct, which was set to 0.25 in order to test only genes present in a minimum fraction of 25% of the cells. This was clarified in the Methods section of the manuscript.

17. Author performed strict quality controls and followed stringent parameters to discard unhealthy or outlier cells. However, author should apply doublet removal algorithm to all samples and confirm that the only singlet cells were used for downstream clustering and analysis. The cell numbers among the 14 samples are quite drastic and its highly likely that samples with very high cell number could have significant number of doublets. Author used CCA Seurat method to integrate different samples.

We applied the doublet removal algorithm DoubletFinder (McGinnis *et al.*, 2019; reference 10 of this point-by-point response) to our samples, and redid the integration workflow in Seurat. This did not result in any relevant differences related to cell type detection or gene expression in fibroblast subpopulations (see figure below). In addition, after removing potential doublets, the dataset contained only 1,044 (0.9%) cells less than before (114,009 cells instead of 115,053) including 220 (0.8%) fibroblasts that were removed (27,162 cells instead of 27,382). We therefore concluded that doublets represent a negligible factor in our study and that reproducing the entire set of analyses after doublet removal is not necessary.

18. Can author perform GSEA analysis on iCAFs and myCAFs population with respective marker to further corroborate the population identity?

The new Supplementary Figures 4d and 4e now include GSEA plots showing an enrichment of the gene expression signatures corresponding to pancreatic cancer-related iCAFs and myCAFs (Öhlund *et al.*, 2017 and Elyada *et al.*, 2019; references 23 & 24 of the manuscript) in the BD- and cSCC-derived iCAFs and myCAFs, respectively.

19. There are a few labelling errors of figure panels throughout the paper.

We have revised all the figures included in the manuscript, main and supplementary, correcting all labeling errors.

Reviewer #3:

1. The similarities of the two main subsets to iCAFs and myCAFs could be substantiated by looking at their similarity scores to published iCAF and myCAF signatures, like the ones in e.g. Elyada, Cancer Discovery, 2019.

We have now calculated similarity scores to pancreatic cancer-related iCAF and myCAF gene signatures (Öhlund *et al.*, 2017 and Elyada *et al.*, 2019; references 23 & 24 of the manuscript). The results show that our iCAFs and myCAFs are similar to the iCAFs and myCAFs observed in pancreatic cancer and have been included in our new Supplementary Figure 4f.

2. Figure 3a, which is used for in situ validation requires staining at the same time with all four markers to allow the conclusion that two subsets (C3+/IGF1+/MMP11-/Wnt5A- and C3-/IGF1/MMP11+/Wnt5A+) have been “in-situ-validated”.

For the multiplexed RNA FISH analyses, all tissue sections were hybridized at the same time with 11 probes capable of detecting expression of all genes of interest, including *C3*, *IGF1*, *MMP11*, and *WNT5A*. However, for the sake of clarity, Figure 3 and Supplementary Figures 5 & 6 (former Supplementary Figure 4) display only the channels that are relevant to illustrate the corresponding results. In the new Supplementary Figures 5a and 5b we have now included a representative microscopy image showing the channels for *C3*, *IGF1*, *MMP11* and *WNT5A*. This new figure complements those that were originally included to validate the existence of iCAFs and myCAFs in BD and cSCC tumors.

3. The in situ analyses (Fig. 3) should be expanded to include some information on differential localization of the two main subsets in relationship to e.g. malignant cells, T-cells, macrophages and vessels.

The multiplexed RNA FISH analyses included in the original manuscript were performed, amongst others, with a probe for *KRT14* to localize healthy and malignant keratinocytes. As requested, new multiplexed RNA FISH experiments were performed with probes for *CD3D*, *AIF1*, *VWF* and *RGS5*, which are specific markers of T cells, macrophages, vascular endothelial cells and pericytes, respectively. These assays did not reveal differential localization of the two main CAF subsets, which is now clarified in the Results section of the manuscript.

4. Some data on how iCAF/myCAF abundance in different cases is associated with clinical characteristics, including survival, would add to the study.

Our scRNA-seq protocol requires the direct processing of clinical samples. As such, no longer-term data, such as survival, is available (and also precluded by the ethical permit of this study).

5. The data from the bioinformatics analyses (Fig. 4) on cell-cell communications suggests a series of in situ analyses where the proposed inter-cellular communications could be validated.

As requested, the revised version of the manuscript also includes multiplexed RNA FISH assays that validate the predicted iCAF- and myCAF-related CXCL12-CXCR4 and WNT5A-MCAM interactions, respectively. The results are now shown in our new Figures 4d and 4e.

Moreover, by using the R package LIANA (Dimitrov *et al.*, 2022; reference 44 of the manuscript), we now provide additional confirmation for the specificity of the predicted iCAF and myCAF interactions with different non-malignant cell types in cSCC tumors. This is now included in the new Supplementary Figure 7e (see also Reviewer #2, point 12).

6. The preliminary indications that the two CAF subsets show different similarities to the four fibroblasts subsets of normal skin (Fig. 1) is an interesting finding that merits further attention.

We agree with the reviewer and have now also included trajectory analyses performed with the scRNA-seq data that further support the development and evolution of CAFs in BD and cSCC patients from fibroblasts present in normal skin. These results are now shown in our new Supplementary Figure 4a. In addition, pairwise comparisons of the most representative genes of iCAFs or myCAFs, and the four subtypes of dermal fibroblasts detected the largest overlaps between pro-inflammatory fibroblasts and iCAFs (15 genes), and between mesenchymal fibroblasts and myCAFs (22 genes). These results are shown in our new Supplementary Figure 4k.

References

1. Reynolds, G. et al. Developmental cell programs are co-opted in inflammatory skin disease. *Science* **371**, 6527 (2021).
2. Inman, G. J. et al. The genomic landscape of cutaneous SCC reveals drivers and a novel azathioprine associated mutational signature. *Nat. Commun.* **9**, 3667 (2018).
3. Salgado, R. et al. CKS1B amplification is a frequent event in cutaneous squamous cell carcinoma with aggressive clinical behaviour. *Genes Chromosomes Cancer* **49**, 1054-1061 (2010).

4. Sekulic, A. et al. Loss of inositol polyphosphate 5-phosphatase is an early event in development of cutaneous squamous cell carcinoma. *Cancer Prev. Res. (Phila.)* **3**, 1277-1283 (2010).
5. Hameetman, L. et al. Molecular profiling of cutaneous squamous cell carcinomas and actinic keratoses from organ transplant recipients. *BMC Cancer* **13**, 58 (2013).
6. Purdie, K. J. et al. Single nucleotide polymorphism array analysis defines a specific genetic fingerprint for well-differentiated cutaneous SCCs. *J. Invest. Dermatol.* **129**, 1562-1568 (2009).
7. Newman, A. M. et al. Determining cell type abundance and expression from bulk tissues with digital cytometry. *Nat. Biotechnol.* **37**, 773-782 (2019).
8. Steen, C. B., Liu, C. L., Alizadeh, A. A., Newman, A. M. Profiling Cell Type Abundance and Expression in Bulk Tissues with CIBERSORTx. *Methods Mol. Biol.* **2117**, 135-157 (2020).
9. Das Mahapatra, K. et al. A comprehensive analysis of coding and non-coding transcriptomic changes in cutaneous squamous cell carcinoma. *Sci. Rep.* **10**, 3637 (2020).
10. McGinnis, C. S., Murrow, L. M., Gartner, Z. J. DoubletFinder: Doublet Detection in Single-Cell RNA Sequencing Data Using Artificial Nearest Neighbors. *Cell Syst.* **8**, 329-337.e324 (2019).

Reviewers' Comments:

Reviewer #1:

Remarks to the Author:

The authors have adequately addressed my comments.

Reviewer #2:

Remarks to the Author:

The authors have done a great job at responding to the reviewer queries.

Only one thing, for all the violin plots in the figures, could the statistical tests and associated significance p values be added to the figures or figure legends?

Reviewer #3:

Remarks to the Author:

Earlier main point 1: Additional analyses (Supp. Fig. 4f) address this question.

Earlier main point 2: Additional analyses (Supp. Fig. 5a and Supp. Fig 5b) does not convincingly demonstrate the expected marker combinations. Furthermore, main Fig. 3 a-c does not well illustrate the expected marker combinations. More specifically

- the two lower right panels of Fig. 3a indicate co-localization of IGF1 and Wnt5A not predicted by the proposed iCAF/myCAF marker pattern.

-the two panels of Fig. 3b show very few cells demonstrating the predicted marker profiles; Col3A1/C3/IGF1-triple positive cells (left panel) or Col3A1/MMP11/Wnt5a-double positive cells (right panel).

Earlier main point 3: No new results regarding spatial enrichment of iCAFs and myCAF are presented, and novelty of study is thus limited.

Earlier main point 4: No new results regarding clinico-pathological associations of the CAF subsets are presented, and novelty of study is thus limited.

Earlier main point 5: Bioinformatics findings regarding cell-cell interactions have been expanded by novel non-quantitative in situ analyses (Fig. 4d and 4e). Functional relevance of interactions should be strengthened by some quantitative data demonstrating iCAF-proximity-dependent T-cell phenotypes or myCAF-proximity- dependent endothelial cell phenotypes.

Earlier main point 6: Some new information has been added partially addressing this point (Supp. Fig. 4a and Supp. Fig. 4k)

New main point: Data do not sufficiently well support the claim of the Title of distinct origin of CAF subsets (also see reviewer 2, main point 5).

Reviewer #2

Only one thing, for all the violin plots in the figures, could the statistical tests and associated significance p values be added to the figures or figure legends?

Statistical tests have been performed for all violin plots included in the manuscript. *P*-values are now provided for all figures, and the main text and the figure legends have been updated accordingly.

Reviewer #3

Earlier main point 2: Additional analyses (Supp. Fig. 5a and Supp. Fig 5b) does not convincingly demonstrate the expected marker combinations.

Furthermore, main Fig. 3 a-c does not well illustrate the expected marker combinations. More specifically

- the two lower right panels of Fig. 3a indicate co-localization of IGF1 and Wnt5A not predicted by the proposed iCAF/myCAF marker pattern.

-the two panels of Fig. 3b show very few cells demonstrating the predicted marker profiles; Col3A1/C3/IGF1-triple positive cells (left panel) or Col3A1/MMP11/Wnt5a-double positive cells (right panel).

As for Supplementary Figures 5a-b, we have now replaced the previous pictures with more magnified versions, indicating iCAFs and myCAFs with differently colored arrows. This reduces the number of cells visible in the panels but allows better visualization of the markers.

As for the new point regarding Figure 3a, please note that cells that appear positive for *IGF1* and *WNT5A* transcripts do not express *COL3A1*, so they are in any case not CAFs. In fact, *COL3A1*-positive CAFs showing co-staining for combinations of iCAF and myCAF marker genes were not taken into account during quantification (Figures 3b and 3d), as no clear subtype classification was possible. Importantly, only 18% of all detected CAFs (*COL3A1*⁺) in cSCC could not be classified as iCAFs or myCAFs due to mixed marker combinations, whereas 72% of all CAFs could be clearly identified as one or another CAF subtype ($p < 0.0001$, t-test). This has now been clarified in the Results and Methods sections of the manuscript, and the additional quantification has been included in the new Supplementary Figure 6a.

As for the new point regarding Fig. 3b, additional quantification was performed and showed that the majority of cSCC-derived iCAFs was defined by co-expression of *COL3A1* and only one of the subtype-specific markers (67%). In comparison, the fraction of triple-positive iCAFs (33%) was significantly smaller ($p < 0.001$, t-test). Similarly, 59% of all cSCC-derived myCAFs were double-positive, whereas only 41% were triple-positive ($p < 0.01$, t-test). This has now been clarified in the Results and Methods sections, and the additional quantification has also been included in the new Supplementary Figure 6b of the manuscript. In Figure 3b, we have now also indicated double- and triple-positive cells with differently colored arrows.

Earlier main point 3: No new results regarding spatial enrichment of iCAFs and myCAF are presented, and novelty of study is thus limited.

As already mentioned in the Discussion section and in the former point-by-point response letter, we did not observe any specific spatial pattern for the observed CAF subpopulations with respect to any malignant and non-malignant cell type(s) in our multiplexed RNA FISH assays. We observed iCAFs and myCAFs throughout the TME (Figure 3a and Supplementary Figures 5c-d), and no highly specialized CAF subtype associated with blood vessels could be identified. While this is different from observations in other cancer types it can be explained by the fact that cSCC-derived CAFs arise from resident dermal fibroblasts (manuscript reference 29). This is now clarified in the Discussion section of the manuscript.

Earlier main point 4: No new results regarding clinico-pathological associations of the CAF subsets are presented, and novelty of study is thus limited.

As discussed in the previous point-by-point response letter, when performing scRNA-seq experiments with fresh clinical samples obtained from patients via surgery, long-term outcome data are not available. Their analysis was therefore always beyond the scope of this study. However, we agree with the reviewer that this is a limitation of the present study, and that it is important that future studies address clinicopathological associations of CAF subpopulations in more detail. This has now been clarified in the Discussion section of the manuscript.

Earlier main point 5: Bioinformatics findings regarding cell-cell interactions have been expanded by novel non-quantitative in situ analyses (Fig. 4d and 4e). Functional relevance of interactions should be strengthened by some quantitative data demonstrating iCAF-proximity-dependent T-cell phenotypes or myCAF-proximity-dependent endothelial cell phenotypes.

As requested, we have now performed quantifications for the two interactions shown in Figures 4d-e. Our analyses show that although both CAF subpopulations were detected throughout the TME, an enrichment of iCAFs (*COL3A1*⁺, *C3*⁺) expressing the ligand *CXCL12* was observed in close proximity to T cells (*CD3D*⁺) expressing the receptor *CXCR4*. Moreover, myCAFs showing co-expression of *COL3A1* and *WNT5A* were enriched next to vascular endothelial cells (*VWF*⁺) expressing the mRNA encoding the MCAM receptor. These new data are now included in Supplementary Figures 8f-g.

New main point: Data do not sufficiently well support the claim of the Title of distinct origin of CAF subsets (also see reviewer 2, main point 5).

We are open to discussing possible modifications to the title with the editor. This being said, Reviewer #2 seems to be satisfied with our response to his/her concerns and does not suggest any changes. We also believe that our data justify the present title: gene expression analyses showed statistically significant upregulation of the previously defined pro-inflammatory fibroblast gene signature (manuscript reference 27) in iCAFs, whereas statistically significant upregulation of mesenchymal fibroblast-related genes was observed in myCAFs (Figure 2d). Similar results were obtained when performing pairwise comparisons with the most representative genes of iCAFs or myCAFs and the four dermal fibroblast subtypes: the largest overlaps were detected between pro-inflammatory fibroblasts and iCAFs (15 genes), and between mesenchymal fibroblasts and myCAFs (22 genes) (Supplementary Figure 4k). Finally, integrated analysis of the transcriptomes of all healthy and diseased skin cells showed that cutaneous iCAFs clustered significantly with pro-inflammatory fibroblasts ($p < 0.0001$, paired t-test), whereas myCAFs clustered significantly with mesenchymal fibroblasts ($p < 0.0001$, paired t-test) (Figure 2e).

Reviewers' Comments:

Reviewer #3:

Remarks to the Author:

Comments have been properly addressed.